# SLASH the Sink: Sharpening Structural Attention Inside LLMs

Yiming Liu [1]  Bin Lu* [1]  Xinbing Wang [1]  Chenghu Zhou [2]  Meng Jin* [1]

## Abstract

Large Language Models (LLMs) show remarkable semantic understanding but often struggle with structural understanding when processing graph topologies in a serialized format. Existing solutions rely on training external graph-based adapters or fine-tuning, which incur high costs and lost generalizability. In this work, we investigate the internal mechanisms of LLMs and present a critical finding: *LLMs spontaneously reconstruct the graph's topology internally*, evidenced by a distinct "sawtooth" pattern in their attention maps that structurally aligns with the "token-level adjacency matrix". However, this intrinsic structural understanding is diluted by the attention sink. We theoretically formalize this dilution as a representation bottleneck, stemming from a fundamental conflict: the model's anisotropic bias, essential for language tasks, suppresses the topology-aware local aggregation required for graph reasoning. To address this, we propose a training-free solution, named **S**tructura**L** **A**ttention **SH**arpening (SLASH), which amplifies this internal structural understanding via a plug-and-play attention redistribution. Experiments on pure graph tasks and molecular prediction validate that SLASH delivers significant and consistent performance gains across diverse LLMs.[1]

## 1. Introduction

Large Language Models (LLMs) have achieved unprecedented success in natural language processing. This has spurred growing interest in applying their power to structured data, particularly graphs, which are ubiquitous in fields ranging from social networks to molecular biology. While promising, effectively enabling LLMs to comprehend and reason over graph topologies remains a significant challenge, leading to a variety of proposed methodologies.

**Prior Works and Challenges.** Current research predominantly follows two paradigms. The first involves *hybrid architectures* that combine GNNs and LLMs (Chen et al., 2024b; Zhao et al., 2023). While effective, this approach introduces architectural complexity and training overhead, often sacrificing the LLM's generalizability (Liu et al., 2024a), precisely the property needed to handle out-of-distribution challenges inherent in graph learning (Lu et al., 2024). The second paradigm involves *feeding serialized graph topologies* directly to LLMs (Kim et al., 2022; Wang et al., 2023). However, these work often treats the LLM as a black box, focusing on input encoding strategies (Fatemi et al., 2024) rather than examining *how* LLMs internally process these flattened structures. This "black box" treatment overlooks a fundamental mechanism: the *attention sink* (Xiao et al., 2024). This phenomenon, where models allocate disproportionate attention to initial tokens to manage information flow in topologically-flat text (Barbero et al., 2025), conflicts when interpreting structured data. The impact of this mechanism on an LLM's ability to reason over graph topologies remains a open question.

**Our Work.** Motivated by the limitations of the "black box" approach, our work pioneers a mechanistic exploration of *how* LLMs process serialized graphs. Our investigation yielded a crucial, two-part discovery. First, we found that LLMs are not structurally blind; they **spontaneously reconstruct the graph's topology internally**, evidenced by a distinct "sawtooth" pattern in their attention maps (Figure 1, left). Second, we observed that this emergent capability is actively suppressed by the attention sink. This observation led us to theorize a fundamental conflict between *semantic anisotropy and topology-aware local aggregation*: the model's pre-trained bias for directional, anisotropic language processing clashes with the local neighborhood aggregation required for graph structural reasoning, creating a representation bottleneck. Based on this diagnosis, we designed SLASH, a solution that works by amplifying the model's latent structural signal rather than injecting external knowledge, as depicted in Figure 1 (right). To summarize,

---

[1]Shanghai Jiao Tong University, Shanghai, China [2]Institute of Geographical Science and Natural Resources Research, Chinese Academy of Sciences, Beijing, China. Correspondence to: Bin Lu <robinlu1209@sjtu.edu.cn>, Meng Jin <jinm@sjtu.edu.cn>.

*Proceedings of the 43$^{rd}$ International Conference on Machine Learning*, Seoul, South Korea. PMLR 306, 2026. Copyright 2026 by the author(s).

[1]The code is available at https://github.com/liuyiming01/SLASH.

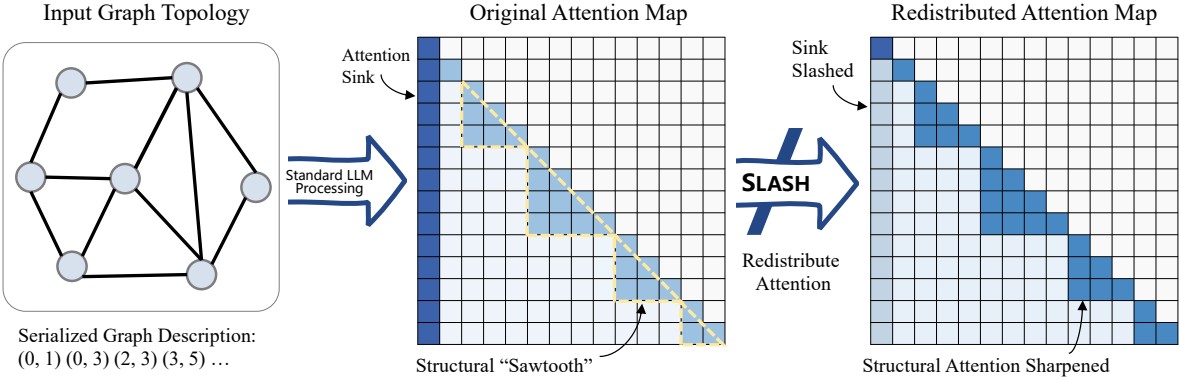

*Figure 1.* The core mechanism of SLASH. While a standard LLM dilutes the latent structural "sawtooth" pattern with a dominant attention sink (left), SLASH sharpens this topological signal, enhancing the model's focus on the internal structure (right).

the main contributions of our work are as follows:

- We identify the conflict between *semantic anisotropy* and *topology-aware local aggregation* as a fundamental bottleneck, providing the first evidence that an LLM's latent ability to reconstruct graph topology is suppressed by the attention sink.

- We propose SLASH, a training-free, plug-and-play methodology that amplifies the model's intrinsic structural understanding at inference time.

- We experimentally validate SLASH's ability to resolve the attention sink bottleneck, showing it consistently and significantly enhances graph reasoning across diverse LLMs.

## 2. Related Work

**LLM for Graph Learning.** The application of LLMs to graph-structured data is a growing research area, with existing approaches largely falling into two main paradigms. (1) *Combining GNNs with LLMs.* This dominant approach combines GNNs for structural encoding with LLMs for semantic understanding. Implementations range from GNN-enhanced LLMs (Chen et al., 2024b; Tang et al., 2024b), to scalable collaborative frameworks (Zhao et al., 2023; Wen & Fang, 2023; Wu et al., 2026), multi-view models (Shirasuna et al., 2024), and LLM-augmented GNNs aiming for generalization (Liu et al., 2024a). (2) *Graph serialization for LLMs.* This line of work explores the LLM's intrinsic ability to process serialized topologies, a concept supported by findings that pure Transformers can be powerful graph learners (Kim et al., 2022). Sparked by inquiries into solving graph problems in text (Wang et al., 2023), this area now includes instruction-tuned models (Wang et al., 2024b; Tang et al., 2024a; Chen et al., 2024a) and architectures with injected structural biases (Wang et al., 2025). Further work explores encoding strategies to better represent

graph structures in a textual format or interpret their internal processing (Fatemi et al., 2024; El et al., 2025). However, the former approach introduces architectural and training overhead, while the latter often fails to examine *how* LLMs internally process flattened graph structures.

**Attention Mechanisms and Sinks.** The self-attention mechanism is fundamental to the Transformer architecture (Vaswani et al., 2017), and research analyzes its mechanisms, from head-level syntactic roles and inactivity to layer-wise linguistic properties and overall depth efficiency (Clark et al., 2019; Skean et al., 2025; Csordás et al., 2025; Sandoval-Segura et al., 2025). A recent discovery is the "attention sink" phenomenon, where LLMs allocate excessive attention to initial tokens (Xiao et al., 2024). Interpretations of its function vary, from preventing representation over-mixing (Barbero et al., 2025) to being an emergent property of information compression (Queipo-de-Llano et al., 2025), while its emergence during pre-training has been empirically linked to the constraints of softmax normalization (Gu et al., 2025). Proposed solutions range from post-hoc, training-free calibration (Yu et al., 2024; Han et al., 2025) to architectural modifications like gated attention designed to be sink-free (Qiu et al., 2025). This body of research, however, is confined to the context of topologically-flat text. Consequently, the impact of these attention phenomena on an LLM's ability to interpret serialized graph structures remains a unexamined and open question.

## 3. Preliminary

In this section, we present the problem formulation of the graph reasoning task and describe its serialization format. We then introduce a structured serialization protocol, which is critical for exposing the latent topological patterns within the model. Finally, we briefly revisit the self-attention mechanism to formally introduce the concept of attention sink.

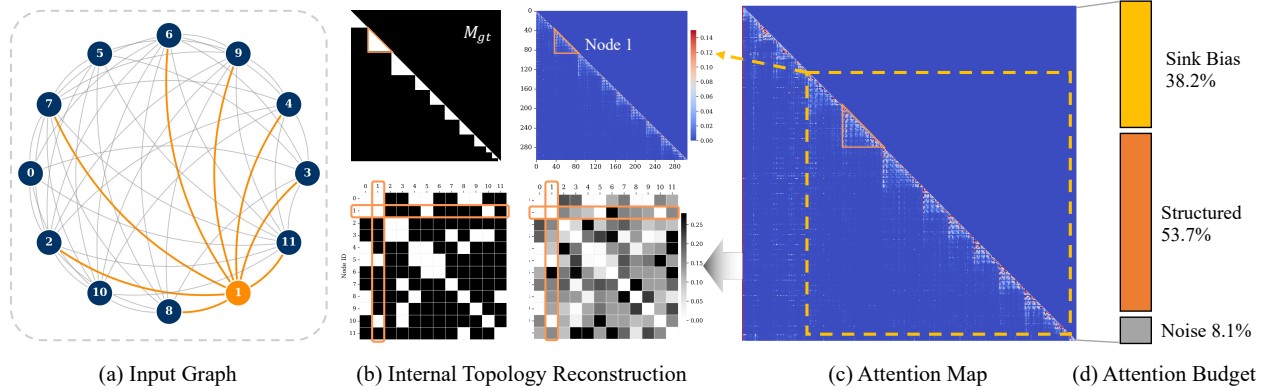

(a) Input Graph    (b) Internal Topology Reconstruction    (c) Attention Map    (d) Attention Budget

*Figure 2.* Mechanistic evidence of internal graph reconstruction. (a) Input graph. (b) Internal Topology Reconstruction: Comparison of ground-truth structures (token-level $M_{gt}$ and node adjacency) with their attention-derived counterparts (sawtooth pattern and reconstructed adjacency). (c) Full attention map. (d) Attention Budget: Proportional breakdown of sink bias, structural aggregation, and noise.

### 3.1. Problem Formulation

**Graph Reasoning Task.** We consider a graph reasoning task where the input consists of a graph $\mathcal{G} = (\mathcal{V}, \mathcal{E})$ and a natural language query $\mathcal{Q}$. The goal of the Large Language Model (LLM) is to generate the optimal response $\mathcal{Y}$ based on the serialized representation of the graph. Formally, the probability of the output is modeled as $P(\mathcal{Y} \mid S(\mathcal{G}), \mathcal{Q})$, where $S(\cdot)$ is the serialization function.

**Graph Serialization Specification.** The serialization function $S(\cdot)$ maps the graph into a textual representation, typically via edge tuples (e.g., `(u,v)`). While standard lists allow arbitrary ordering, we introduce a **Source-Node Aggregation** protocol requiring edges from the same source to appear contiguously. Specifically, a sequence like `(0,1)(2,3)(0,3)` is rearranged to `(0,1)(0,3)(2,3)`. Although LLMs are robust to edge permutations, this aggregation aligns physical adjacency with local topology. Such alignment provides the spatial basis to make the internal topological patterns observable.

### 3.2. Self-Attention and Attention Sink

For the $h$-th attention head in the $l$-th layer, the self-attention mechanism computes an attention matrix $\mathbf{A}^{(l,h)}$ where, due to the Softmax function, scores in each row are normalized to sum to one: $\sum_{j=1}^{i} \mathbf{A}_{i,j}^{(l,h)} = 1$.

Latest work shows that LLMs assign significant attention scores to specific token positions (especially initial tokens), termed the "Attention Sink" (Xiao et al., 2024). This is understood to arise from the Softmax constraint itself (Gu et al., 2025) and helps preserve representation distinctiveness in text (Barbero et al., 2025). While essential for topologically-flat text, we argue this reflects a pre-training bias that degrades the model's graph reasoning capabilities. Understanding this degradation is a central challenge of this work.

## 4. Observation and Theoretical Analysis

This section presents our core theoretical analysis. We first reveal that LLMs internally reconstruct graph topology, then formalize this mechanism to identify a fundamental limitation—the *representation bottleneck*. The analysis concludes by deriving the attention sharpening principle as the theoretical solution.

### 4.1. Empirical Observation: LLMs Internally Reconstruct Topology

To probe how LLMs process serialized graphs, we analyze their internal attention maps. Our analysis uncovers a latent, highly structured phenomenon, as illustrated in Figure 2. Given a input graph (Fig. 2a), the full attention map of specific intermediate heads reveals a distinct "sawtooth" pattern emerging amidst the dominant attention sink (Fig. 2c).

This pattern is not random. To formalize its structure, we define a binary, token-level adjacency matrix $\mathbf{M}_{gt}$, where $\mathbf{M}_{gt}[i,j] = 1$ if tokens $x_i$ and $x_j$ (with $j \leq i$) belong to edge descriptions from the same source node. As shown in Figure 2b, the observed sawtooth pattern is structurally aligned with the ground-truth matrix $\mathbf{M}_{gt}$. This provides direct evidence that the LLM is spontaneously reconstructing the graph topology from the serialized text.

This is not an isolated finding. The phenomenon is consistently observed across various models (e.g., Llama-3.1, Qwen3), concentrated in specific intermediate layers, as detailed in Appendix J. However, as quantified by the attention budget breakdown (Fig. 2d), this emergent structural signal is heavily diluted by the sink. This observation motivates our central hypothesis: a latent structural understanding exists within LLMs, but it is suppressed rather than absent.

## 4.2. LLM Layers as Implicit Causal GATs

Our observation in Sec. 4.1 suggests that specific LLM layers perform a locally-focused, topology-aware computation. To formalize this, we analyze the self-attention update for a token's output representation $\mathbf{h}_i$, computed as $\mathbf{h}_i = \sum_{j=0}^{i} \alpha_{i,j} \mathbf{v}_j$, where $\alpha_{i,j}$ is the attention weight and $\mathbf{v}_j$ is the value vector.

Guided by the observed "sawtooth" pattern, which aligns with the causal adjacency matrix $\mathbf{M}_{gt}$, we decompose this update into three functional components:

$$\mathbf{h}_i = \underbrace{\alpha_{i,0} \mathbf{v}_0}_{\text{Sink Bias}} + \underbrace{\sum_{j \in \mathcal{N}_i^{\text{pre}}} \alpha_{i,j} \mathbf{v}_j}_{\text{Structural Aggregation}} + \underbrace{\sum_{k \notin \mathcal{N}_i^{\text{pre}} \cup \{0\}} \alpha_{i,k} \mathbf{v}_k}_{\text{Noise } (\epsilon_i)} \quad (1)$$

Here, $\mathcal{N}_i^{\text{pre}} = \{j \mid 1 \le j < i, \mathbf{M}_{gt}[i,j] = 1\}$ is the set of the token's preceding graph neighbors. This decomposition reveals the layer acts as an implicit causal Graph Attention Network (GAT), featuring a dominant sink bias and a topological aggregation term over local neighbors.

As quantified by the attention budget breakdown in Figure 2d, the residual term ($\epsilon_i$) carries a small fraction of the total attention mass (see Appendix D for empirical noise ratios across models and tasks). We approximate it away to expose the dominant competition between sink bias and structural aggregation, yielding the simplified model:

$$\mathbf{h}_i \approx \alpha_{i,0} \mathbf{v}_0 + \sum_{j \in \mathcal{N}_i^{\text{pre}}} \alpha_{i,j} \mathbf{v}_j \quad (2)$$

This simplified formulation highlights a key conflict. Unlike a standard GAT where attention is normalized only over local neighbors (i.e., $\sum_{j \in \mathcal{N}_i} \alpha_{i,j} = 1$), the LLM's global Softmax constraint forces $\alpha_{i,0} + \sum_{j \in \mathcal{N}_i^{\text{pre}}} \alpha_{i,j} \approx 1$. This creates a zero-sum competition for attention, which we attribute to the LLM's pre-training inertia: a learned preference for a strong attention sink on flat text is misapplied to serialized graph data. This mechanism creates the bottleneck that we analyze next.

## 4.3. The Representation Bottleneck

This subsection analyzes the representation bottleneck caused by a fundamental conflict between the LLM's pre-training bias and the structural nature of graphs. We first explain the conceptual origin of this conflict and then provide its formal mathematical analysis.

**Conceptual Origin: Semantic Anisotropy vs. Topology-Aware Local Aggregation.** The representation bottleneck stems from a conflict between two opposing processing requirements.

- **Semantic Anisotropy in LLMs.** LLM representations for natural language are anisotropic, occupying a narrow cone in the embedding space (Ethayarajh, 2019). This directional semantic structure is effective for text, where tokens possess rich, self-contained semantics. The attention sink is understood as a mechanism to preserve this property by preventing representation over-mixing (Barbero et al., 2025).

- **Topology-Aware Local Aggregation for Graphs.** In contrast, graph nodes are often semantically vacuous; their identity is not self-contained but constructed relationally through local neighborhood aggregation, where a node's representation is built by aggregating information from its graph neighbors. This process is the foundation of standard GNNs such as MPNNs and GATs.

- **The Conflict.** A conflict emerges when the LLM applies its pre-training bias to graph data. The strong directional semantic bias in token representations competes with the local neighborhood aggregation needed for structural reasoning. When this topological signal is suppressed, structurally distinct nodes fail to acquire unique representations, collapsing into a narrow cone and losing the identity-defining information encoded in the topology.

**Mathematical Analysis.** We now formalize the representation bottleneck by analyzing how the sink's attention budget, $\lambda_i = \alpha_{i,0}$, degrades structural information at both node and graph levels. We begin by reformulating our simplified model from Eq. (2):

$$\mathbf{h}_i = \lambda_i \mathbf{v}_0 + (1 - \lambda_i) \mathbf{h}_i^{\text{topo}}, \quad (3)$$

where $\mathbf{h}_i^{\text{topo}} = \frac{1}{1-\lambda_i} \sum_{j \in \mathcal{N}_i^{\text{pre}}} \alpha_{i,j} \mathbf{v}_j$ is the pure topological representation, normalized to behave like a standard GAT aggregation.

At the node level, the sink's influence causes a geometric contraction in the embedding space.

**Theorem 4.1** (Geometric Contraction). *The Euclidean distance between two node representations $\mathbf{h}_k$ and $\mathbf{h}_l$ is contracted by a factor of $(1 - \lambda)$ compared to their sink-free distance, assuming a uniform sink weight $\lambda_k \approx \lambda_l \approx \lambda$:*

$$\|\mathbf{h}_k - \mathbf{h}_l\| = (1 - \lambda)\|\mathbf{h}_k^{topo} - \mathbf{h}_l^{topo}\|. \quad (4)$$

This result (proof in Appendix A.1) proves that as the sink's budget $\lambda$ increases, structurally distinct nodes become geometrically indistinguishable.

At the graph level, this node-level contraction culminates in a more damaging effect: the suppression of structural information in the spectral domain. From a Graph Signal

Processing (GSP) perspective, graph structure understanding relies on identifying local variations, which are encoded in high-frequency components. We measure this with the graph's Dirichlet Energy, $E_{\text{Dir}}(\mathbf{H}) = \text{tr}(\mathbf{H}^\top \mathbf{L} \mathbf{H})$, where $\mathbf{L}$ is the graph Laplacian.

**Proposition 4.2** (Dirichlet Energy Decay). *The sink's presence with weight $\lambda$ quadratically suppresses the Dirichlet Energy:*

$$E_{Dir}(\mathbf{H}) \approx (1 - \lambda)^2 E_{Dir}(\mathbf{H}^{topo}). \qquad (5)$$

This decay (proof in Appendix A.2) occurs because the sink term is filtered out by the Laplacian, leaving only the quadratically scaled topological term. Together, these results reveal that the attention sink acts as an aggressive spectral low-pass filter, forcing distinct node representations to collapse and suppressing the high-frequency signals that encode structural differences.

### 4.4. Reversing the Representation Bottleneck

Based on our analysis, the solution is to reverse the representation bottleneck by re-allocating attention from the sink to the topological structure. We introduce a control factor $\gamma \in [0, 1)$ and define the attention sharpening principle which redistributes the attention budget recovered from the sink, $(1 - \gamma)\lambda_i$, proportionally among all non-sink tokens:

$$\alpha'_{i,j} = \begin{cases} \gamma \cdot \alpha_{i,j} & \text{if } j = 0 \\ \alpha_{i,j} \cdot \left(1 + \frac{(1-\gamma)\alpha_{i,0}}{1-\alpha_{i,0}}\right) & \text{if } j > 0 \end{cases} \qquad (6)$$

We now show this principle reverses the bottleneck at both node and graph levels.

**Node-Level Expansion.** At the node level, sharpening reverses the geometric contraction.

**Theorem 4.3** (Geometric Expansion). *The sharpening principle induces a scaling of the Euclidean distance between any two node representations. The scaling factor, which we term the Topological Amplification Ratio, $\rho(\lambda, \gamma)$, is given by:*

$$\frac{\|\mathbf{h}'_k - \mathbf{h}'_l\|}{\|\mathbf{h}_k - \mathbf{h}_l\|} = \rho(\lambda, \gamma) = \frac{1 - \gamma\lambda}{1 - \lambda}. \qquad (7)$$

This result (proof in Appendix A.3) confirms that our method directly counteracts the geometric indistinguishability caused by the sink.

**Graph-Level Spectral Amplification.** At the graph level, this geometric expansion manifests as a spectral amplification.

**Proposition 4.4** (Spectral Amplification). *The sharpened Dirichlet Energy, $E'_{Dir}(\mathbf{H})$, can be approximated as the original energy scaled by the square of the Topological*

*Amplification Ratio, $\rho^2$:*

$$E'_{Dir}(\mathbf{H}) \approx \rho(\lambda, \gamma)^2 E_{Dir}(\mathbf{H}) = \left(\frac{1 - \gamma\lambda}{1 - \lambda}\right)^2 E_{Dir}(\mathbf{H}). \qquad (8)$$

This operation (derivation in Appendix A.4) acts as a controllable high-pass amplifier, recovering the high-frequency structural components suppressed by the sink and effectively "de-blurring" the graph representation.

**Principled Selection of $\gamma$.** The control factor $\gamma$ governs the trade-off between structural amplification and model fidelity. On one hand, setting $\gamma \to 0$ (Hard Removal) maximizes topological amplification but risks what has been termed "representation collapse" (Barbero et al., 2025) by ignoring the sink's stabilizing role. On the other hand, setting $\gamma \to 1$ (No Intervention) preserves maximum fidelity but fails to correct the representation bottleneck.

Therefore, an optimal $\gamma \in (0, 1)$ exists as a necessary compromise. Instead of being a fixed empirical value, its selection is guided by a lightweight, automated calibration process. We determine the optimal $\gamma$ for each model by evaluating performance on a small, disjoint calibration dataset across a predefined range of values. This principled approach ensures that we enhance structural clarity while respecting the model's foundational semantic space.

## 5. Methodology

Guided by our theoretical analysis, we propose **S**tructura**L** **A**ttention **SH**arpening (SLASH), a training-free solution that enhances the intrinsic structural understanding of LLMs. SLASH consists of two main phases: an **Offline Phase** for identification and calibration, and an **Online Phase** to apply attention sharpening during inference.

### 5.1. Offline Phase: Identification and Calibration

The offline phase operates on a small sample of graph data to identify topology-aware heads ($\mathcal{H}_{topo}$) and calibrate the optimal sharpening factor ($\gamma$). As a critical first step, each graph's edge list in this data undergoes the **Source-Node Aggregation** protocol (Sec. 3). This provides the necessary spatial basis for revealing the internal topological patterns required by the subsequent screening process.

**Entropy-based Activity Filtering.** Inspired by prior studies (Jawahar et al., 2019), our first stage distinguishes active from inactive heads using Matrix-Based Entropy, defined as:

$$H(\mathbf{A}^{(l,h)}) = -\sum_j p_j \log p_j, \quad \text{where } p_j = \frac{\sigma_j^2}{\|\mathbf{A}^{(l,h)}\|_F^2}, \qquad (9)$$

and $\sigma_j$ denotes the $j$-th singular value of $\mathbf{A}^{(l,h)}$. A high score indicates a complex attention pattern, and these active heads are concentrated in the model's intermediate layers (see Appendix B for an example with Llama-3.1-8B). We retain this pool of high-entropy heads, isolated via an automatic threshold, for the next stage of analysis.

**Structural Feature Extraction.** To evaluate the structural alignment of these candidate heads, we extract features from each attention map $\mathbf{A}^{(l,h)}$: (1) *Region Extraction*, to isolate token regions corresponding to edge descriptions; (2) *Binarization*, where we retain only the top-$k$ attention scores (with $k$ being the number of non-zero elements in the ground-truth adjacency matrix $\mathbf{M}_{gt}$) to create a binary matrix $\mathbf{B}^{(l,h)}$; and (3) *Morphological Smoothing*, where a morphological closing (one binary dilation followed by one binary erosion with a $3 \times 3$ square structuring element) is applied to $\mathbf{B}^{(l,h)}$ to connect fragmented regions and yield a smoothed map $\hat{\mathbf{B}}^{(l,h)}$.

**Attention Concentration Scoring.** With the smoothed feature map, we define an attention concentration score $\mathcal{C}$ to quantify alignment with the ground-truth topology $\mathbf{M}_{gt}$. The score measures concentration within a target region $\Omega_{in}$ versus leakage into an outer region $\Omega_{out}$, based on an in-region error $\mathcal{E}_{in}$ and an out-of-region error $\mathcal{E}_{out}$:

$$\mathcal{E}_{in} = \frac{1}{|\Omega_{in}|} \sum_{(i,j) \in \Omega_{in}} (1 - \hat{\mathbf{B}}_{ij}^{(l,h)}), \qquad (10)$$

$$\mathcal{E}_{out} = \frac{1}{|\Omega_{out}|} \sum_{(i,j) \in \Omega_{out}} \hat{\mathbf{B}}_{ij}^{(l,h)}. \qquad (11)$$

The final score is computed as $\mathcal{C}^{(l,h)} = (1 - \mathcal{E}_{in}) \cdot (1 - \mathcal{E}_{out})$.

**Head Selection via Automatic Thresholding.** The final set of topology-aware heads, $\mathcal{H}_{topo}$, is selected by applying Otsu's method (Otsu, 1979) to find thresholds for both the entropy scores and the concentration scores: $\mathcal{H}_{topo} = \{(l, h) \mid H(\mathbf{A}^{(l,h)}) \geq T_S \text{ and } \mathcal{C}^{(l,h)} \geq T_C\}$.

**Sharpening Factor Calibration.** Following head identification, we determine the optimal sharpening factor $\gamma$ for each model-task pair via a lightweight, automated process. We evaluate performance on a small, disjoint calibration dataset across a range of $\gamma$ values (e.g., $[0.1, \ldots, 1.0]$) and select the best-performing one for that specific configuration. This one-time calibration for each pair yields a dedicated $\gamma$ that is used for all of its corresponding test evaluations.

### 5.2. Online Phase: Inference-Time Sharpening

We implement the online phase as a plug-and-play module that intercepts the attention map of each targeted head $(l, h) \in \mathcal{H}_{topo}$ during inference. It then applies the sharpening principle (Eq. (6)) using the pre-calibrated $\gamma$ to redistribute attention. This intervention modifies only the

attention scores, enabling the model to leverage its latent structural understanding for the downstream task.

## 6. Experiment

In this section, we conduct a comprehensive set of experiments to evaluate the effectiveness of SLASH. The evaluation aims to answer the following research questions:

- **RQ1:** How effective is SLASH at enhancing the performance of both general and fine-tuned LLMs on diverse graph reasoning tasks?

- **RQ2:** What is the impact of key hyperparameters and intervention strategies on SLASH's effectiveness?

- **RQ3:** How do the different components of the offline identification method contribute to its effectiveness?

- **RQ4:** What do case studies reveal about the effectiveness and boundary conditions of SLASH?

### 6.1. Experimental Setup

**Datasets.** We evaluate SLASH on two benchmark suites. **(1) GraphInstruct**(Chen et al., 2024a) assesses performance on pure graph computational tasks. **(2) MolecularNet**(Wu et al., 2018) provides 5 datasets for real-world Molecular Property Prediction (MPP). For MolecularNet, we convert SMILES strings into graph descriptions (atoms as nodes, bonds as edges) to expose the explicit topology. Prompt details appear in Appendix H.

**Models and Baselines.** Our evaluation includes both general-purpose and domain-specific LLMs. General-purpose models include the Llama3 and Qwen3 series (Team, 2024; Yang et al., 2025), covering various model sizes. Domain-specific baselines include state-of-the-art fine-tuned models: GraphWiz variants (Mistral-7B-RFT, LLaMA2-7B/13B-DPO) for GraphInstruct, and MolecularGPT (Liu et al., 2024b) for MolecularNet.

**Implementation Details.** The sharpening factor $\gamma$ is calibrated independently for each model and task via the process in Sec.5.1. All main results adopt the layer-level intervention strategy (Sec.6.3). Appendix C provides further details on the computational environment and overhead.

### 6.2. Main Results (RQ1)

To demonstrate the effectiveness of SLASH, we present the main results on both pure graph tasks (GraphInstruct) and real-world property prediction (MolecularNet).

**Performance on GraphInstruct.** Table 1 shows that SLASH significantly improves the performance of general-purpose LLMs. In contrast, it provides only marginal gains

*Table 1.* Accuracy on the GraphInstruct benchmark. SLASH significantly improves the performance of general-purpose LLMs, while providing only marginal gains for fine-tuned models.

| Model | Method | Cycle | Connect | Bipartite | Topology | Shortest | Triangle | Flow | Hamilton | Subgraph | Avg |
|---|---|---|---|---|---|---|---|---|---|---|---|
| Llama-3.2-3B | Vanilla | 0.180 | 0.340 | 0.005 | 0.045 | 0.030 | 0.000 | 0.075 | 0.092 | 0.000 | 0.085 |
| | SLASH | **0.573** | 0.340 | **0.235** | **0.048** | 0.030 | **0.048** | **0.113** | **0.278** | **0.268** | **0.214** |
| Llama-3.1-8B | Vanilla | 0.660 | 0.730 | 0.235 | 0.060 | 0.018 | 0.018 | 0.128 | 0.305 | 0.385 | 0.282 |
| | SLASH | **0.680** | **0.797** | **0.547** | 0.022 | 0.015 | **0.052** | **0.175** | **0.315** | **0.560** | **0.352** |
| Qwen3-4B | Vanilla | 0.645 | 0.420 | 0.048 | 0.003 | 0.018 | 0.000 | 0.000 | 0.223 | 0.005 | 0.151 |
| | SLASH | **0.690** | 0.388 | **0.195** | 0.003 | **0.025** | 0.003 | **0.015** | **0.290** | **0.145** | **0.195** |
| Qwen3-8B | Vanilla | 0.395 | 0.537 | 0.177 | 0.000 | 0.000 | 0.000 | 0.062 | 0.177 | 0.215 | 0.174 |
| | SLASH | **0.905** | **0.858** | **0.570** | 0.000 | 0.000 | 0.000 | 0.062 | **0.318** | **0.588** | **0.367** |
| Qwen3-14B | Vanilla | 0.545 | 0.217 | 0.155 | 0.018 | 0.025 | 0.013 | 0.068 | 0.190 | 0.588 | 0.202 |
| | SLASH | **0.740** | **0.733** | 0.155 | **0.100** | **0.037** | 0.013 | **0.070** | **0.247** | 0.588 | **0.298** |
| GraphWiz-Mistral-7B | Vanilla | 0.923 | 0.897 | 0.726 | 0.200 | 0.306 | 0.389 | 0.306 | 0.291 | 0.863 | 0.544 |
| | SLASH | 0.914 | 0.891 | 0.726 | **0.211** | 0.306 | 0.369 | **0.311** | **0.300** | **0.866** | 0.544 |
| GraphWiz-LLaMA2-7B | Vanilla | 0.874 | 0.814 | 0.854 | 0.471 | 0.231 | 0.523 | 0.437 | 0.814 | 0.754 | 0.642 |
| | SLASH | **0.891** | 0.806 | 0.854 | 0.446 | 0.231 | 0.517 | 0.437 | 0.809 | **0.757** | 0.639 |
| GraphWiz-LLaMA2-13B | Vanilla | 0.891 | 0.891 | 0.883 | 0.554 | 0.246 | 0.546 | 0.423 | 0.494 | 0.809 | 0.637 |
| | SLASH | **0.931** | **0.894** | 0.883 | 0.554 | **0.249** | 0.450 | **0.443** | **0.620** | 0.754 | **0.642** |

*Table 2.* Accuracy on the MolecularNet benchmark. SLASH enhances the performance of general-purpose LLMs, while the domain-specialized MolecularGPT shows minimal change.

| Model | Method | BACE | BBBP | ClinTox | HIV | Tox21 |
|---|---|---|---|---|---|---|
| Llama-3.2-3B | Vanilla | 0.085 | 0.495 | 0.675 | 0.932 | 0.418 |
| | SLASH | **0.493** | 0.495 | **0.793** | 0.932 | **0.500** |
| Llama-3.1-8B | Vanilla | 0.335 | 0.430 | 0.875 | 0.932 | 0.510 |
| | SLASH | 0.335 | **0.560** | 0.875 | 0.932 | **0.518** |
| Qwen3-4B | Vanilla | 0.230 | 0.315 | 0.578 | 0.113 | 0.488 |
| | SLASH | **0.525** | **0.420** | **0.843** | **0.935** | 0.488 |
| Qwen3-8B | Vanilla | 0.233 | 0.060 | 0.018 | 0.003 | 0.003 |
| | SLASH | **0.280** | **0.430** | **0.855** | **0.670** | **0.485** |
| Qwen3-14B | Vanilla | 0.345 | 0.430 | 0.875 | 0.932 | 0.513 |
| | SLASH | **0.615** | 0.430 | 0.875 | 0.932 | 0.497 |
| MolecularGPT | Vanilla | 0.650 | 0.430 | 0.200 | 0.068 | 0.470 |
| | SLASH | 0.650 | 0.430 | 0.200 | 0.068 | **0.480** |

for models already fine-tuned on this domain. We suggest this is because the fine-tuning process itself optimizes structural attention, achieving a topological alignment similar to what SLASH induces at inference-time. Consequently, our training-free method offers limited additional benefit.

**Performance on MolecularNet.** Table 2 shows this trend extends to MolecularNet, where SLASH consistently enhances the performance of general-purpose LLMs on molecular property prediction. Consistent with our hypothesis, the gain is minimal for the domain-specialized MolecularGPT. This result validates that SLASH is effective for real-world prediction tasks by sharpening the model's understanding of explicit graph topology.

## 6.3. Hyperparameter and Strategy Analysis (RQ2)

We now analyze the key design choices for SLASH: the sharpening factor $\gamma$, the intervention granularity, and the input aggregation protocol.

**Sensitivity to Sharpening Factor $\gamma$.** Figure 3 illustrates performance sensitivity to the sharpening factor $\gamma$ on representative tasks; we provide comprehensive results across all benchmarks in Appendix I. General-purpose LLMs exhibit model-specific sensitivity, with performance trends varying across models. This confirms that an optimal $\gamma \in (0, 1)$ exists and validates our calibration strategy. In contrast, the performance of the fine-tuned GraphWiz model remains stable, which is consistent with our hypothesis that fine-tuning has already optimized the model's structural attention.

**Head-level vs. Layer-level Intervention.** We compare two intervention strategies: applying SLASH to identified heads ($\mathcal{H}_{\text{topo}}$), or to all heads within the layers containing them. Table 3 shows layer-level intervention consistently provides superior and more stable performance, whereas the granular head-level approach can degrade it. We hypothesize this is because a Transformer layer acts as a holistic computational unit. A head-level intervention disrupts the collaborative mechanism between heads, compromising the layer's overall information flow, leading to sub-optimal performance.

**Input Aggregation Analysis.** Table 4 evaluates the impact of the Source-Node Aggregation protocol at inference time. The results yield two insights. First, aggregation improves performance for the baseline and SLASH-enhanced models, indicating that a structured input simplifies the task. Second, more critically, SLASH remains effective on non-aggregated

*Table 3.* Head-level vs. Layer-level Intervention. Layer-level intervention provides consistent gains, whereas the non-uniform, head-level approach can be unstable and degrade performance.

| Model | Method | Cyc. | Bip. | Ham. | BBBP | Tox21 |
|-------|--------|------|------|------|------|-------|
| Llama-3.1-8B | Vanilla | 0.660 | 0.235 | 0.305 | 0.430 | 0.510 |
| | SLASH (Head) | 0.670 | 0.325 | **0.318** | 0.430 | 0.513 |
| | SLASH (Layer) | **0.680** | **0.547** | 0.315 | **0.560** | **0.518** |
| Qwen3-8B | Vanilla | 0.395 | 0.177 | 0.177 | 0.060 | 0.003 |
| | SLASH (Head) | 0.143 | 0.098 | 0.175 | 0.003 | 0.000 |
| | SLASH (Layer) | **0.905** | **0.570** | **0.318** | **0.430** | **0.485** |

*Table 4.* Impact of Source-Node Aggregation at inference. SLASH provides robust performance gains regardless of edge ordering.

| Model | Agg. | Method | Cyc. | Bip. | Ham. | BBBP | Tox21 |
|-------|------|--------|------|------|------|------|-------|
| Llama-3.1-8B | w/o | Vanilla | 0.565 | 0.243 | 0.298 | 0.430 | 0.513 |
| | | SLASH | **0.668** | **0.548** | **0.320** | **0.625** | **0.520** |
| | with | Vanilla | 0.663 | 0.293 | 0.295 | 0.430 | 0.510 |
| | | SLASH | **0.680** | **0.630** | **0.318** | 0.430 | **0.523** |
| Qwen3-8B | w/o | Vanilla | 0.315 | 0.175 | 0.155 | 0.023 | 0.000 |
| | | SLASH | **0.843** | **0.593** | **0.318** | **0.430** | **0.488** |
| | with | Vanilla | 0.380 | 0.190 | 0.193 | 0.060 | 0.005 |
| | | SLASH | **0.898** | **0.535** | **0.318** | **0.430** | **0.498** |

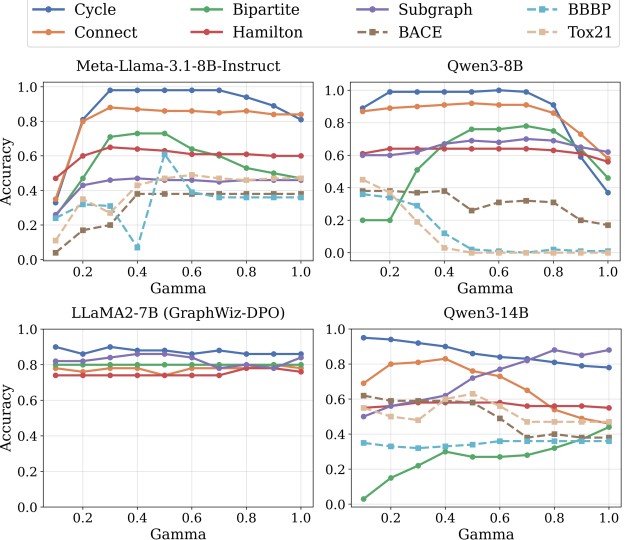

*Figure 3.* Performance sensitivity to the sharpening factor $\gamma$. General-purpose LLMs show model-specific sensitivity, while the fine-tuned model's performance remains stable.

*Table 5.* Ablation of identification components within the layer-level strategy. The full method, combining both entropy and concentration filtering, provides the most robust performance.

| Model | Method | Cyc. | Bip. | Ham. | BBBP | Tox21 |
|-------|--------|------|------|------|------|-------|
| Llama-3.1-8B | All Layers | 0.675 | 0.008 | 0.068 | **0.570** | 0.513 |
| | Entropy-only | 0.640 | 0.288 | 0.243 | 0.540 | 0.503 |
| | Struct-only | 0.678 | 0.480 | 0.010 | 0.430 | 0.513 |
| | SLASH (Ours) | **0.680** | **0.547** | **0.315** | 0.560 | **0.518** |
| Qwen3-8B | All Layers | 0.810 | **0.638** | 0.318 | 0.330 | 0.195 |
| | Entropy-only | 0.410 | 0.175 | 0.175 | 0.415 | 0.188 |
| | Struct-only | **0.918** | 0.603 | 0.318 | 0.420 | 0.303 |
| | SLASH (Ours) | 0.905 | 0.570 | **0.318** | **0.430** | **0.485** |
| Qwen3-14B | All Layers | 0.545 | 0.138 | 0.230 | 0.423 | 0.510 |
| | Entropy-only | 0.628 | **0.168** | 0.218 | 0.430 | **0.513** |
| | Struct-only | 0.693 | 0.105 | 0.243 | 0.113 | 0.510 |
| | SLASH (Ours) | **0.740** | 0.155 | **0.247** | **0.430** | 0.497 |

inputs. This demonstrates that the heads identified offline are fundamental to structural reasoning, not mere detectors of a specific pattern induced by aggregation. Our offline protocol uses aggregation as a proxy to find these heads, but the online sharpening is robust to input permutation because it amplifies the intrinsic structural signal. We additionally report fine-tuned model results under non-aggregated inputs in Appendix F, which show a consistent pattern.

### 6.4. Ablation Study (RQ3)

To validate our two-stage identification method, we conduct an ablation study within the superior layer-level strategy. We compare our full SLASH method against three variants: one targeting all layers (All Layers), one using only the entropy criterion (Entropy-only), and one using only the attention concentration criterion (Struct-only). Table 5 shows the full method provides the most robust performance. This result validates our design, as each component addresses a critical failure mode: entropy filtering alone selects active but structurally-unfocused heads, while concentration filtering alone risks selecting inactive ones. The combination is therefore essential to identify layers containing computationally active and topologically aligned heads.

### 6.5. Case Studies and Limitations (RQ4)

Figure 4 illustrates how SLASH corrects reasoning failures. The vanilla model fabricates a path in its textual response to incorrectly answer 'Yes'. In contrast, SLASH provides the correct 'No' answer. This demonstrates that our method

prevents structural hallucinations by enhancing the model's topological understanding.

We also examine specific limitations to understand where SLASH struggles. On the Graph-SST2 (Yuan et al., 2023) sentiment task, where syntax trees form the graph structure, SLASH yields unstable results: it degrades Llama-3.1-8B's performance ($0.638 \rightarrow 0.583$) while improving Qwen3-8B's ($0.513 \rightarrow 0.613$). We attribute this to the task's nature, where sentiment depends on word semantics, not syntactic topology. Therefore, amplifying structural signals can be counterproductive when the graph structure is irrelevant to the task, delineating the scope of SLASH's applicability.

Complementarily, Appendix G evaluates SLASH on broader graph settings—node classification, link prediction, and

graph QA—where consistent improvements confirm its generalizability to tasks in which topological structure drives the reasoning process.

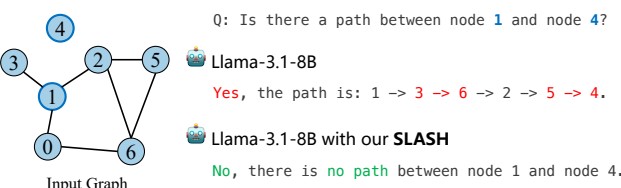

*Figure 4.* Case study on connectivity detection. The vanilla model incorrectly answers 'Yes' by generating a hallucinated path. SLASH prevents hallucinations, ensuring a grounded 'No'.

# 7. Conclusion

In this paper, we reveal LLMs' latent ability to reconstruct graph topology, identifying the conflict between semantic anisotropy and topology-aware local aggregation as the bottleneck. We propose SLASH, a training-free attention mechanism that resolves this bottleneck, achieving significant performance gains on both pure graph and molecular prediction tasks. For future work, we will investigate incorporating this principle into model training to further improve LLMs' understanding of structured data and study the theoretical underpinnings of the conflict.

# Acknowledgements

This work was supported by National Natural Science Foundation of China (No. T2421002, 62602003, 62272293) and Postdoctoral Fellowship Program of CPSF (No. GZB20250806).

# Impact Statement

This paper presents work whose goal is to advance the field of Machine Learning. There are many potential societal consequences of our work, none which we feel must be specifically highlighted here.

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

# A. Omitted Proofs

### A.1. Proof of Theorem 4.1

*Proof.* Let $\mathbf{h}_k$ and $\mathbf{h}_l$ be the final representations for two nodes, and $\mathbf{h}_k^{\text{topo}}$, $\mathbf{h}_l^{\text{topo}}$ be their pure topological representations. Assuming a uniform sink weight $\lambda_k \approx \lambda_l \approx \lambda$, we start from Eq. (3):

$$\mathbf{h}_k = \lambda\mathbf{v}_0 + (1 - \lambda)\mathbf{h}_k^{\text{topo}}$$
$$\mathbf{h}_l = \lambda\mathbf{v}_0 + (1 - \lambda)\mathbf{h}_l^{\text{topo}}$$

Subtracting the two equations gives:

$$\mathbf{h}_k - \mathbf{h}_l = (\lambda\mathbf{v}_0 + (1 - \lambda)\mathbf{h}_k^{\text{topo}}) - (\lambda\mathbf{v}_0 + (1 - \lambda)\mathbf{h}_l^{\text{topo}}) = (1 - \lambda)(\mathbf{h}_k^{\text{topo}} - \mathbf{h}_l^{\text{topo}}).$$

Taking the Euclidean norm of both sides, we get:

$$\|\mathbf{h}_k - \mathbf{h}_l\| = \|(1 - \lambda)(\mathbf{h}_k^{\text{topo}} - \mathbf{h}_l^{\text{topo}})\| = (1 - \lambda)\|\mathbf{h}_k^{\text{topo}} - \mathbf{h}_l^{\text{topo}}\|.$$

$\square$

### A.2. Proof of Proposition 4.2

*Proof.* The Dirichlet Energy of the system is defined as $E_{\text{Dir}}(\mathbf{H}) = \text{tr}(\mathbf{H}^\top \mathbf{L}\mathbf{H})$. In matrix form, the simplified model from Eq. (3) can be written as:

$$\mathbf{H} \approx (1 - \lambda)\mathbf{H}^{\text{topo}} + \lambda\mathbf{1}\mathbf{v}_0^\top,$$

where $\mathbf{1}$ is a column vector of ones. We substitute this into the energy function:

$$E_{\text{Dir}}(\mathbf{H}) \approx \text{tr}\left(\left((1 - \lambda)\mathbf{H}^{\text{topo}} + \lambda\mathbf{1}\mathbf{v}_0^\top\right)^\top \mathbf{L}\left((1 - \lambda)\mathbf{H}^{\text{topo}} + \lambda\mathbf{1}\mathbf{v}_0^\top\right)\right).$$

A fundamental property of the graph Laplacian $\mathbf{L}$ is that it has a null space spanned by the vector of all ones, meaning $\mathbf{L}\mathbf{1} = \mathbf{0}$. Consequently, the cross-terms and the sink-only term involving $\mathbf{L}(\mathbf{1}\mathbf{v}_0^\top)$ become zero. The expression simplifies to:

$$E_{\text{Dir}}(\mathbf{H}) \approx \text{tr}\left(\left((1 - \lambda)\mathbf{H}^{\text{topo}}\right)^\top \mathbf{L}\left((1 - \lambda)\mathbf{H}^{\text{topo}}\right)\right).$$

Since $(1 - \lambda)$ is a scalar, we can factor it out:

$$E_{\text{Dir}}(\mathbf{H}) \approx (1 - \lambda)^2 \text{tr}\left((\mathbf{H}^{\text{topo}})^\top \mathbf{L}\mathbf{H}^{\text{topo}}\right) = (1 - \lambda)^2 E_{\text{Dir}}(\mathbf{H}^{\text{topo}}).$$

$\square$

### A.3. Proof of Theorem 4.3

*Proof.* The new representation $\mathbf{h}_i'$ under sharpening with factor $\gamma$ has a new sink weight $\lambda_i' = \gamma\lambda_i$. Following Eq. (3), this gives $\mathbf{h}_i' = \gamma\lambda_i\mathbf{v}_0 + (1 - \gamma\lambda_i)\mathbf{h}_i^{\text{topo}}$. The distance between two nodes becomes:

$$\|\mathbf{h}_k' - \mathbf{h}_l'\| = (1 - \gamma\lambda)\|\mathbf{h}_k^{\text{topo}} - \mathbf{h}_l^{\text{topo}}\|.$$

From Theorem 4.1, we know $\|\mathbf{h}_k - \mathbf{h}_l\| = (1 - \lambda)\|\mathbf{h}_k^{\text{topo}} - \mathbf{h}_l^{\text{topo}}\|$. The ratio of the new distance to the old distance is therefore:

$$\frac{\|\mathbf{h}_k' - \mathbf{h}_l'\|}{\|\mathbf{h}_k - \mathbf{h}_l\|} = \frac{(1 - \gamma\lambda)\|\mathbf{h}_k^{\text{topo}} - \mathbf{h}_l^{\text{topo}}\|}{(1 - \lambda)\|\mathbf{h}_k^{\text{topo}} - \mathbf{h}_l^{\text{topo}}\|} = \frac{1 - \gamma\lambda}{1 - \lambda}.$$

$\square$

## A.4. Derivation of Proposition 4.4

*Proof.* The new representation matrix is $\mathbf{H}' \approx (1-\gamma\lambda)\mathbf{H}^{\text{topo}} + \gamma\lambda\mathbf{1}\mathbf{v}_0^\top$. Its Dirichlet Energy is $E'_{\text{Dir}}(\mathbf{H}) = \text{tr}((\mathbf{H}')^\top \mathbf{L}\mathbf{H}')$. As established in the proof of Prop. 4.2, the sink term is in the null space of $\mathbf{L}$. Thus, the energy becomes:

$$E'_{\text{Dir}}(\mathbf{H}) \approx (1-\gamma\lambda)^2 E_{\text{Dir}}(\mathbf{H}^{\text{topo}}).$$

From Prop. 4.2, we have $E_{\text{Dir}}(\mathbf{H}) \approx (1-\lambda)^2 E_{\text{Dir}}(\mathbf{H}^{\text{topo}})$, which implies $E_{\text{Dir}}(\mathbf{H}^{\text{topo}}) \approx \frac{E_{\text{Dir}}(\mathbf{H})}{(1-\lambda)^2}$. Substituting this back, we get:

$$E'_{\text{Dir}}(\mathbf{H}) \approx (1-\gamma\lambda)^2 \frac{E_{\text{Dir}}(\mathbf{H})}{(1-\lambda)^2} = \left(\frac{1-\gamma\lambda}{1-\lambda}\right)^2 E_{\text{Dir}}(\mathbf{H}).$$

□

# B. LLM Attention Entropy Distribution

*Figure 5.* Entropy distribution in Llama-3.1-8B. Active heads (high entropy) are concentrated in intermediate layers. (Left) Per-head entropy heatmap with active heads boxed. (Right) Layer-averaged entropy plot, where an automatic threshold (dashed line) isolates the active peak.

# C. Computational Environment and Overhead

**Computational Environment.** Experiments were conducted on a server with eight NVIDIA RTX 4090 GPUs, using PyTorch 2.3.1 and the Hugging Face Transformers library (v4.41.3).

**Overhead Analysis.** As a training-free method, SLASH introduces zero additional parameters. Its computational overhead is minimal:

- **Offline Phase Overhead:** This one-time, per-model cost is dominated by the $\gamma$ calibration, which requires 10 inference runs on a small calibration set (100 samples). The head identification process adds a cost roughly equivalent to a single inference run on the same data.

- **Online Phase Overhead:** The arithmetic of the intervention itself is highly efficient, adding negligible latency per token. However, our implementation requires access to the full attention matrix and is thus not compatible with standard acceleration kernels like FlashAttention.

To quantify this cost, we measured peak GPU memory using Meta-Llama-3.1-8B-Instruct in 4-bit quantization under the same inference setup, as shown in Table 6.

Table 6. Peak GPU memory (MB) under different attention implementations.

| Setting | Tokens | Eager Peak | SLASH Eager Peak | FlashAttn2 Peak |
|---|---|---|---|---|
| Real sample (39 nodes) | 2416 | 15855.5 | 15856.7 | 12161.3 |
| Synthetic (100 nodes) | 6942 | 36111.6 | 36111.7 | 13312.7 |

The sharpening rule itself adds virtually no overhead beyond standard eager attention. The remaining gap between eager and FlashAttention modes is inherent to full attention-map materialization and is not specific to our method.

## D. Empirical Analysis of the Residual Term

To support the approximation in Eq. (2), we report the empirical noise ratio (the fraction of attention mass in the residual term) across models and tasks, as shown in Table 7.

Table 7. Noise ratios across models and tasks.

| Model | Cyc. | Bip. | Ham. | BBBP | Tox21 |
|---|---|---|---|---|---|
| Llama-3.1-8B | 0.106 | 0.097 | 0.114 | 0.111 | 0.110 |
| Qwen3-8B | 0.189 | 0.174 | 0.197 | 0.218 | 0.209 |

We also tested the effect of removing the residual term at inference time on the Cycle task, as reported in Table 8.

Table 8. Impact of removing the residual term at inference.

| Model | Vanilla | SLASH | SLASH (remove noise) |
|---|---|---|---|
| Llama-3.1-8B | 0.660 | 0.680 | 0.686 |
| Qwen3-8B | 0.395 | 0.905 | 0.838 |

These results confirm that the residual term is consistently small, supporting the approximation in Eq. (2). Explicitly removing it at inference does not always improve performance, indicating that it carries some useful signal despite its small magnitude.

## E. Head Selection: Top-$k$ Binarization vs. Direct Soft Alignment

We compare our proposed top-$k$ binarization with morphological closing against a direct soft-alignment alternative for topology-aware head selection. Results are shown in Table 9.

Direct soft alignment can have boundary failures: for example, an attention map with large values mainly on the diagonal can still score highly because many high-value entries fall inside the support of $\mathbf{M}_{gt}$, while not exhibiting the global sawtooth pattern. The proposed pipeline treats the attention map as an image-like object, providing more robust structural identification.

## F. Fine-tuned Models under Non-Aggregated Inputs

To complement the non-aggregated analysis in Table 4 (which covers general-purpose LLMs), we additionally evaluated fine-tuned graph models under non-aggregated (w/o Agg.) inputs, as shown in Table 10.

These results show that fine-tuned models remain mostly neutral to mildly helpful under non-aggregated inputs, confirming that Source-Node Aggregation is an offline diagnostic device rather than a required online assumption.

*Table 9.* Comparison of head selection methods.

| Model | Method | Cyc. | Bip. | Ham. | BBBP | Tox21 |
|-------|--------|------|------|------|------|-------|
| Llama-3.1-8B | Directly Soft | 0.655 | 0.353 | 0.243 | 0.410 | 0.508 |
| | Top-$k$ & Closing | 0.680 | 0.547 | 0.315 | 0.560 | 0.518 |
| Qwen3-8B | Directly Soft | 0.413 | 0.170 | 0.185 | 0.370 | 0.093 |
| | Top-$k$ & Closing | 0.905 | 0.570 | 0.318 | 0.430 | 0.485 |

*Table 10.* Fine-tuned models under non-aggregated inputs.

| Model | Task | Vanilla (w/o Agg.) | SLASH (w/o Agg.) |
|-------|------|--------------------|--------------------|
| GraphWiz-LLaMA2-7B | Cyc. | 0.857 | 0.871 |
| | Bip. | 0.860 | 0.860 |
| | Ham. | 0.606 | 0.609 |
| GraphWiz-Mistral-7B | Cyc. | 0.903 | 0.894 |
| | Bip. | 0.729 | 0.729 |
| | Ham. | 0.257 | 0.274 |
| MolecularGPT | BBBP | 0.430 | 0.430 |
| | Tox21 | 0.490 | 0.505 |

## G. Extended Evaluation on Broader Graph Tasks

To probe the applicability of SLASH beyond the primary benchmarks, we conduct preliminary evaluations on three additional task families using InstructGraph(Wang et al., 2024a) test data (~100 examples per task). For heterogeneous graphs, we serialize the relation as an edge label in the form $(i \rightarrow j, r)$. Results are reported in Table 11.

*Table 11.* Extended evaluation on broader graph tasks.

| Task | Dataset | Model | Vanilla | SLASH |
|------|---------|-------|---------|-------|
| Node Classification | Cora | Llama-3.1-8B | 0.900 | **0.914** |
| | | Qwen3-8B | 0.329 | **0.600** |
| Link Prediction | Wikidata5M | Llama-3.1-8B | 0.230 | **0.260** |
| | | Qwen3-8B | 0.210 | 0.210 |
| Graph QA | PathQuestion | Llama-3.1-8B | 0.329 | **0.357** |
| | | Qwen3-8B | 0.357 | **0.386** |

These results provide preliminary evidence that SLASH can also be useful in broader graph settings. The improvements are consistent with the paper's central perspective: SLASH tends to be helpful when topological structure plays an important role in the reasoning process.

## H. Prompt Formulation for MolecularNet

This section provides the detailed prompt formulations used for the molecular property prediction tasks on the MolecularNet benchmark.

## I. Sensitivity Analysis of Sharpening Factor $\gamma$

## J. Empirical Observation

---

**Prompt for the BACE Task**

---

You are an expert chemist. Your task is to predict the property of a molecule based on its graph structure representation.
- [i, w] means node i has atomic number w (e.g., 6 for Carbon, 8 for Oxygen).
- (i, j) means node i and node j are connected by a chemical bond.
Consider molecular weight, atom count, bond types, and functional groups to assess the compound's drug-likeness and potential as a therapeutic agent for Alzheimer's disease.
Given the graph structure of a molecule, predict its molecular properties by analyzing whether it can inhibit (Yes) or cannot inhibit (No) the Beta-site Amyloid Precursor Protein Cleaving Enzyme 1 (BACE1).
Molecular Graph: `<Molecular Graph Description>`
Q: Can this molecule inhibit BACE1? Answer with only "Yes" or "No".
A:

---

**Prompt for the BBBP Task**

---

You are an expert chemist. Your task is to predict the property of a molecule based on its graph structure representation.
- [i, w] means node i has atomic number w (e.g., 6 for Carbon, 8 for Oxygen).
- (i, j) means node i and node j are connected by a chemical bond.
Given the graph structure of a molecule, predict whether it can penetrate the blood-brain barrier (Yes) or not (No).
Molecular Graph: `<Molecular Graph Description>`
Q: Can this molecule penetrate the blood-brain barrier? Answer with only "Yes" or "No".
A:

---

**Prompt for the ClinTox Task**

---

You are an expert chemist. Your task is to predict the property of a molecule based on its graph structure representation.
- [i, w] means node i has atomic number w (e.g., 6 for Carbon, 8 for Oxygen).
- (i, j) means node i and node j are connected by a chemical bond.
Given the graph structure of a molecule, predict whether it is clinically trial-toxic (Yes) or not (No).
Molecular Graph: `<Molecular Graph Description>`
Q: Is this molecule clinically trial toxic? Answer with only "Yes" or "No".
A:

---

**Prompt for the HIV Task**

---

You are an expert chemist. Your task is to predict the property of a molecule based on its graph structure representation.
- [i, w] means node i has atomic number w (e.g., 6 for Carbon, 8 for Oxygen).
- (i, j) means node i and node j are connected by a chemical bond.
Given the graph structure of a molecule, predict whether a molecule, based on its properties and HIV activity test results, can effectively inhibit HIV replication (Yes) or not (No).
Molecular Graph: `<Molecular Graph Description>`
Q: Can this molecule effectively inhibit HIV replication? Answer with only "Yes" or "No".
A:

---

**Prompt for the Tox21 Task**

---

You are an expert chemist. Your task is to predict the property of a molecule based on its graph structure representation.
- [i, w] means node i has atomic number w (e.g., 6 for Carbon, 8 for Oxygen).
- (i, j) means node i and node j are connected by a chemical bond.
Given the graph structure of a molecule, predict whether it is toxic (Yes) or not toxic (No).
Molecular Graph: `<Molecular Graph Description>`
Q: Is this molecule toxic? Answer with only "Yes" or "No".
A:

---

*Figure 6.* Sensitivity analysis of $\gamma$ on the GraphInstruct dataset.

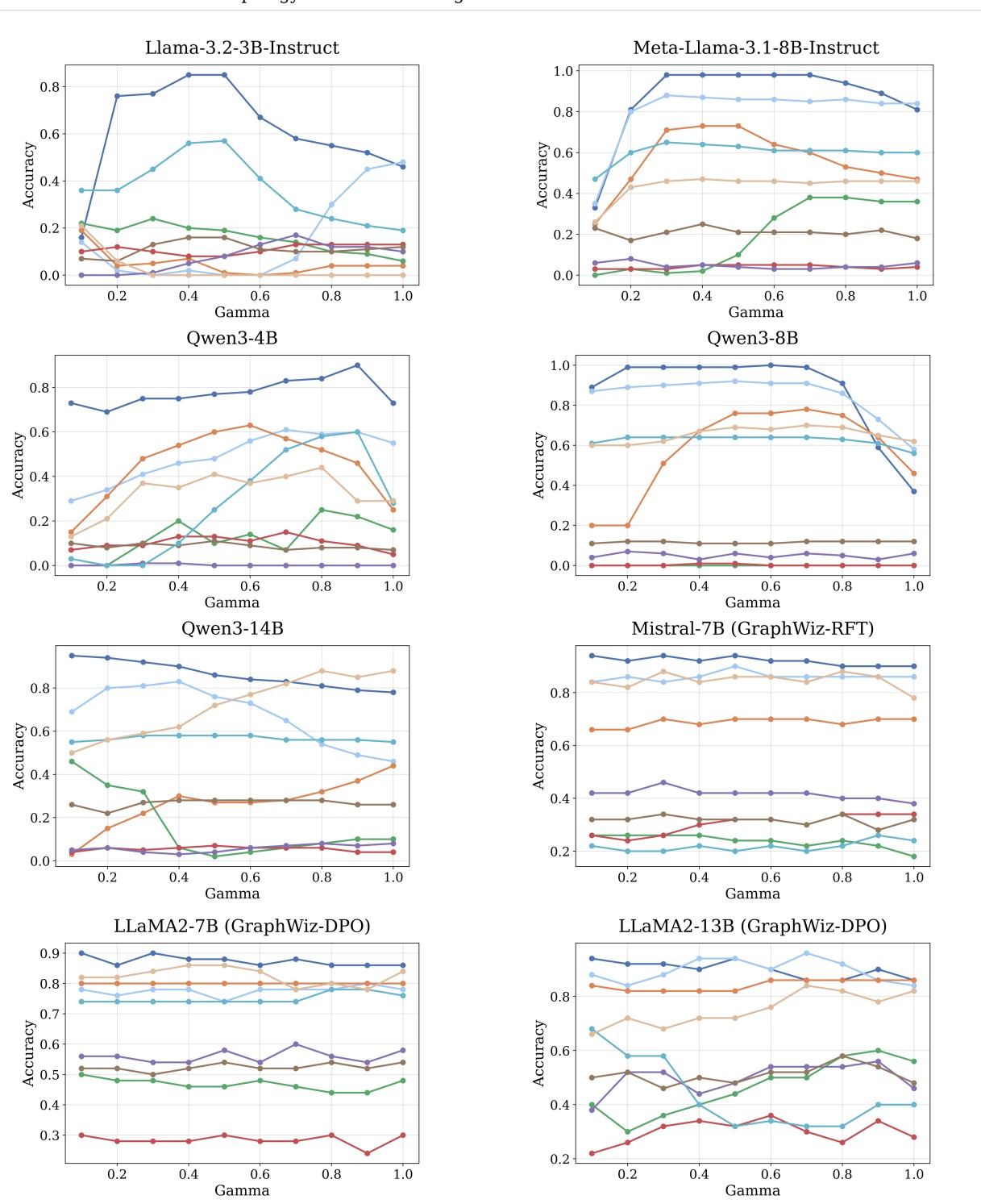

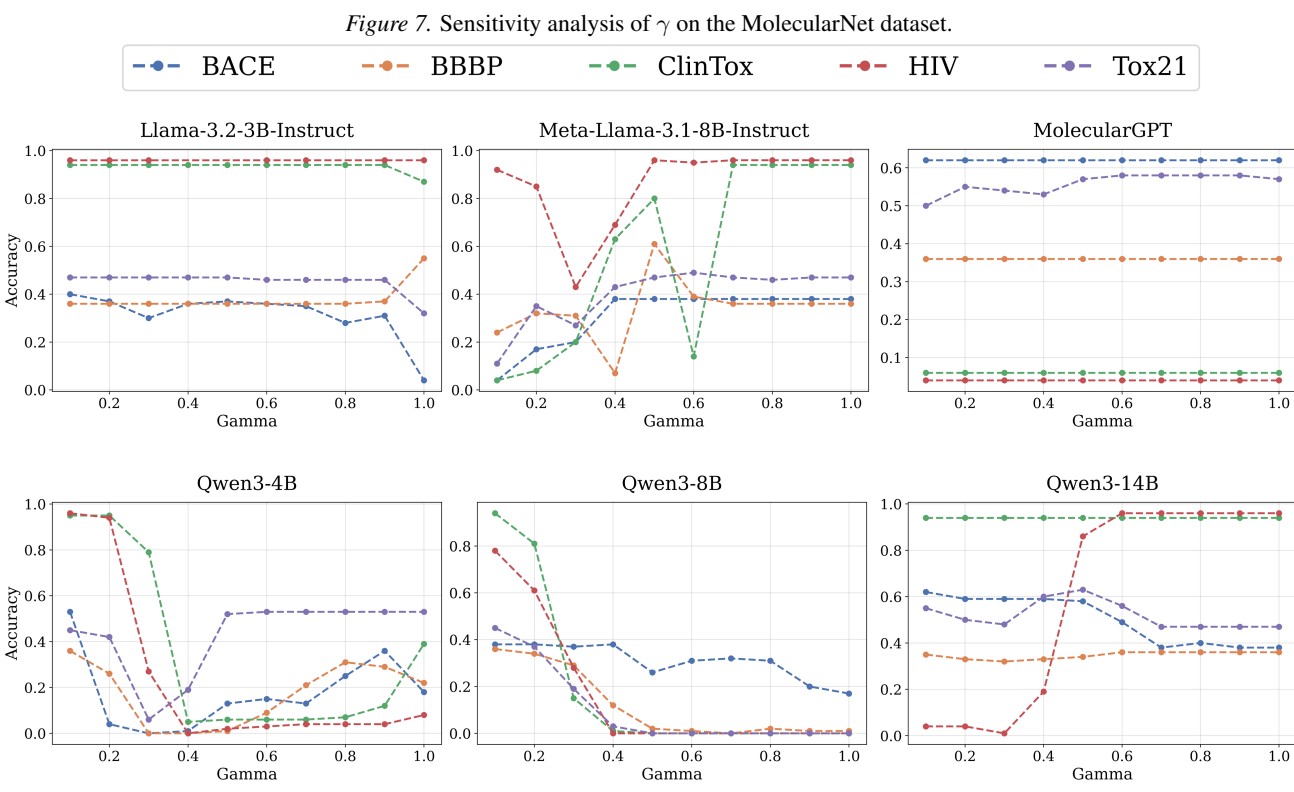

*Figure 7.* Sensitivity analysis of $\gamma$ on the MolecularNet dataset.

*Figure 8.* The attention map of Llama-3.2-3B.

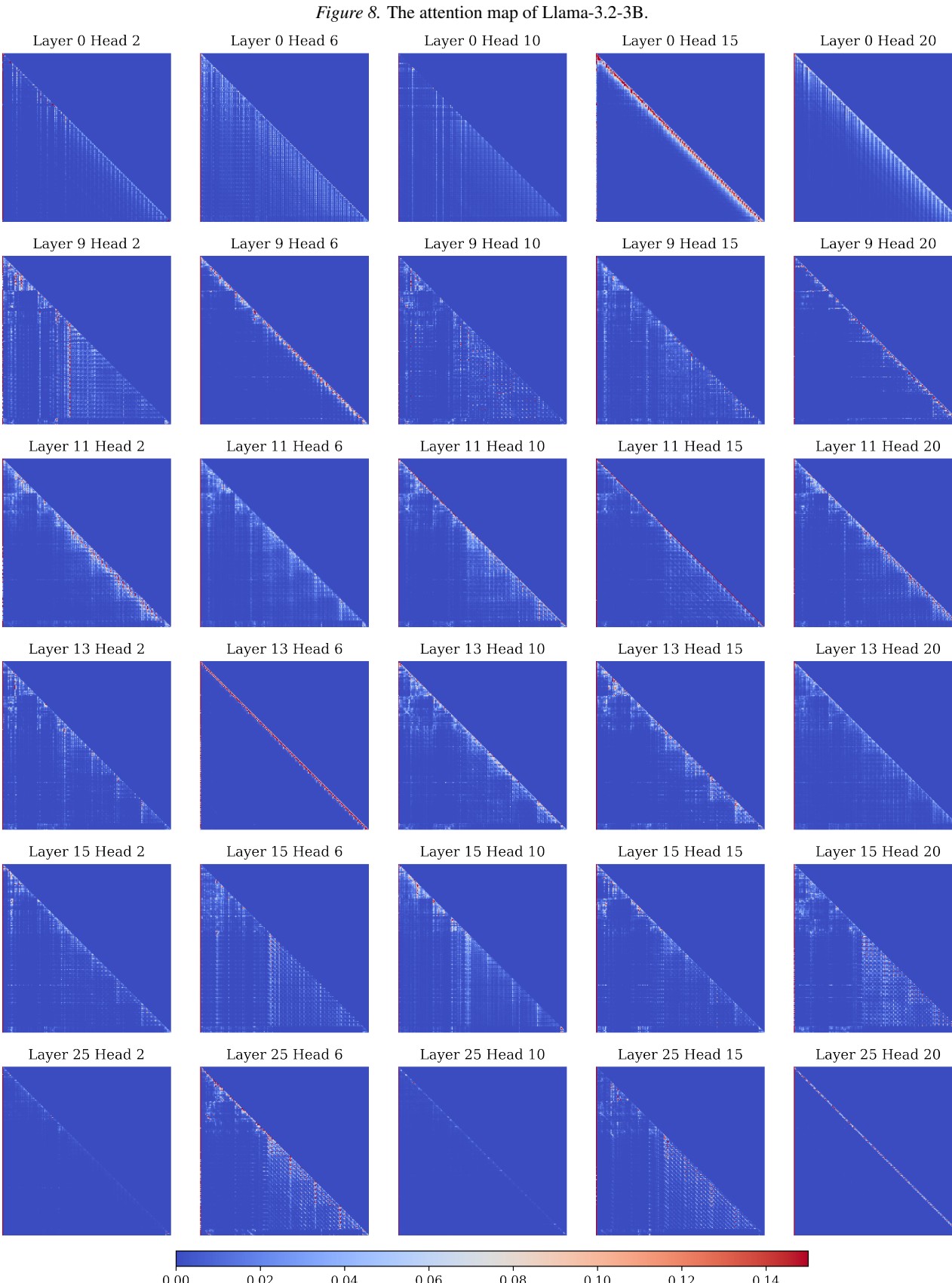

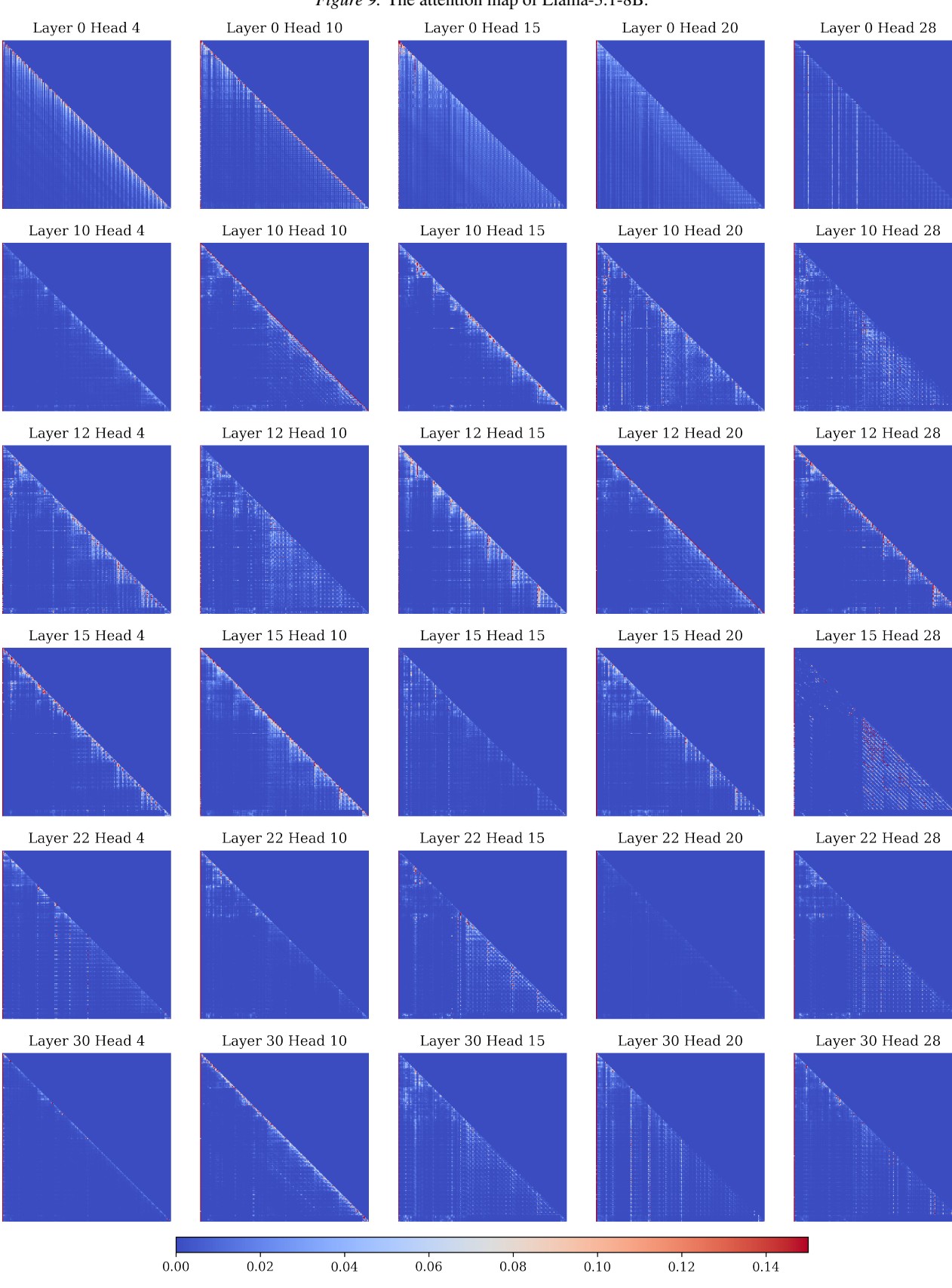

*Figure 9.* The attention map of Llama-3.1-8B.

*Figure 10.* The attention map of Qwen3-4B.

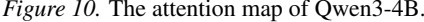

*Figure 11.* The attention map of Qwen3-8B.

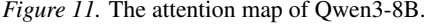

*Figure 12.* The attention map of Qwen3-14B.

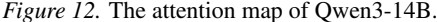

