# OpenReview forum: "SLASH the Sink: Sharpening Structural Attention Inside LLMs"
_ICML.cc/2026/Conference — ICML 2026 regular_

### Official Review · Reviewer_PHFv · 2026-02-24

**Soundness:** 2
**Presentation:** 2
**Significance:** 3
**Originality:** 3
**Overall Recommendation:** 4
**Confidence:** 3

**Summary:**

This paper studies why general-purpose LLMs often struggle to reason over **graph topologies provided as serialized text** (e.g., edge lists). The authors report an empirical observation that, in certain intermediate layers/heads, LLM attention maps exhibit a structured "sawtooth" pattern that aligns with a **token-level adjacency matrix** induced by a "source-node aggregation" serialization protocol.

The paper argues that this latent structural signal is **diluted by attention sinks** (disproportionate attention to early tokens), and formalizes the effect as a representation bottleneck framed as an "anisotropy–isotropy conflict" between language-pretraining geometry and graph diffusion requirements.

Based on this diagnosis, the authors propose **SLASH**, a **training-free, plug-and-play inference-time intervention** that redistributes attention mass away from sink positions to non-sink tokens via a single control parameter $\gamma$. SLASH includes an offline phase to (i) identify "topology-aware" layers/heads via entropy + topology-concentration heuristics and (ii) calibrate $\gamma$ per model/task, and an online phase that applies attention redistribution during generation. The method is evaluated on **GraphInstruct** graph-computation tasks and **MolecularNet** molecular property prediction (converted into explicit graph descriptions), showing accuracy improvements for several general LLMs and smaller gains for domain-tuned baselines.

**Compliance With Llm Reviewing Policy:**

Affirmed.

**Final Justification:**

The paper’s main strengths are its originality and its effort to provide a mechanism-level account of how LLMs process serialized graph topology, and the rebuttal substantially improved my confidence in the work by clarifying the technical choices and addressing most of my soundness and clarity concerns. My remaining concern is about scope and generalization beyond serialized explicit-topology settings, but overall the rebuttal moved my evaluation in a more positive direction and supports a final recommendation on the positive side.

**Key Questions For Authors:**

1. **Clarification on $\sigma_j$ / SVD and morphological closing:**
   In Eq. (9), $\sigma_j$ is not defined—does it refer to the j-th singular value? Could you also clarify where/why SVD is computed and what it is intended to capture? Finally, please specify the exact morphological closing settings used in the pipeline.

2. **Structural feature extraction rationale (top-k binarization & closing):**
   The pipeline binarizes attention by keeping top-k entries (with k matched to the number of non-zeros in $M_{gt}$) and then applies morphological closing. Could you explain the motivation for top-k binarization, and why alternatives such as using $M_{gt}$ directly or a soft alignment metric were not adopted? An ablation on these design choices would strengthen confidence in the head-selection procedure.

3. **On the "noise is negligible" assumption:**
   You refer to the residual term as "noise" and assume it is negligible to derive Eq. (2). Could you provide evidence that this term is consistently small across layers/tasks/models? Also, if it is truly negligible, have you tried removing it at inference time, and what is the impact on performance?

4. **Generality: evaluation on widely used Graph+LLM benchmarks/tasks:**
   Many recent top-venue Graph+LLM papers evaluate on standard real-world graph tasks such as node classification/link prediction on OGB or citation graphs, as well as heterogeneous graphs and graph QA (e.g., LLaGA, GraphGPT, HiGPT, InstructGraph). Given SLASH is an inference-time attention intervention, could you comment on why these benchmark families were not included, and whether there are technical obstacles to applying SLASH in those settings? Even results on a small representative subset would help clarify the scope of applicability.

**Limitations:**

No. The paper would benefit from a more explicit discussion of (i) deployment constraints due to requiring full attention-matrix access (limiting compatibility with efficient attention implementations), and (ii) scope limitations, as performance can degrade on tasks where structural reasoning is not central.

**Strengths And Weaknesses:**

### Strengths
* **Novel perspective and creative idea:** The paper offers a fresh mechanistic perspective on why LLMs struggle with serialized graph topology, and the core framing (linking attention sinks to suppressed topology-aligned patterns) is inventive.
* **Clear mechanism + simple intervention:** The paper connects attention sinks to suppressed topology-aligned attention patterns ("sawtooth") and proposes a clean, training-free attention redistribution rule with a single knob $\gamma$.
* **Promising gains on graph computation tasks:** On GraphInstruct, SLASH substantially improves several general LLMs, while leaving GraphWiz (graph-tuned) mostly unchanged, which is consistent with the stated hypothesis.

### Weaknesses
* **Lack of technical clarity in key steps:** Eq. (9) does not define $\sigma_j$ (the implementation suggests it may be the j-th singular value), and it is unclear where/why SVD is applied and what it is intended to measure; the top-k binarization and morphological closing steps also omit some details.
* **Theory relies on strong simplifications:** The sink/structure/noise decomposition assumes the residual "noise" term is negligible; more empirical evidence across layers/tasks/models would be needed to support the approximation and terminology.
* **Limited scope vs. common Graph+LLM benchmarks:** Experiments focus on GraphInstruct + MoleculeNet only; the paper does not evaluate on widely used graph+LLM settings (e.g., node/link tasks on OGB/citation graphs, heterogeneous graphs, graph QA/captioning), making generality unclear.

---

> ### Author Rebuttal · Authors · 2026-03-31
>
> Dear Reviewer PHFv,
>
> Thank you for the careful reading, encouraging view of our novelty, and precise questions on technical clarity, the noise approximation, and scope. These concerns are very helpful, and we will incorporate the clarifications and additional results below in the revision.
>
> **Q1/W1: Clarification on $\sigma_j$ / SVD and morphological closing.** Thank you for pointing this out. Eq. (9) uses matrix entropy from the normalized singular-value spectrum, where $\sigma_j$ denotes the $j$-th singular value of the attention matrix. We compute SVD on each candidate attention map and use this entropy to measure head activity rather than structure directly: low-entropy heads are often inactive, while higher-entropy heads are more likely to process serialized topology. We therefore use it as a first-stage activity filter before structural scoring. For the preprocessing step, we keep the top-k entries to form a binary mask, with $k$ matched to the number of non-zero entries in $M_{gt}$, and then apply morphological closing to smooth fragmented local regions and expose the global coarse pattern. In our implementation, this closing is one binary dilation followed by one binary erosion with a fixed $3\times3$ square structuring element.
>
> **Q2: Top-k binarization + morphological closing.** Our goal is to identify heads with a genuine **sawtooth-shaped structural pattern**, not merely high entrywise overlap with $M_{gt}$. Direct or soft alignment has boundary failures: for example, an attention map with large values mainly on the diagonal can still score highly because many high-value entries fall inside the support of $M_{gt}$, while not showing the global sawtooth pattern. We therefore treat the attention map as an image-like object: top-k binarization suppresses dispersed small noise and keeps salient structure, while morphological closing smooths local irregularities and exposes the global coarse pattern. This is simpler and more robust than designing many boundary-specific rules for direct alignment. We also compared this design against a direct soft-alignment alternative:
> |Model|Method|Cyc.|Bip.|Ham.|BBBP|Tox21|
> |-|-|-|-|-|-|-|
> |Llama-3.1-8B|Directly Soft|0.655|0.353|0.243|0.410|0.508|
> | |Top-k & Closing|**0.680**|**0.547**|**0.315**|**0.560**|**0.518**|
> |Qwen3-8B|Directly Soft|0.413|0.170|0.185|0.370|0.093|
> | |Top-k & Closing|**0.905**|**0.570**|**0.318**|**0.430**|**0.485**|
>
> These results support the proposed preprocessing.
>
> **Q3/W2: On the “noise is negligible” assumption.** We agree that “negligible” is too strong. A better description is a **small residual mass**. Empirically, the corresponding noise ratio (the fraction of attention mass in the residual term) is relatively stable across tasks/models:
> |Model|Cyc.|Bip.|Ham.|BBBP|Tox21|
> |-|-|-|-|-|-|
> |Llama-3.1-8B|0.106|0.097|0.114|0.111|0.110|
> |Qwen3-8B|0.189|0.174|0.197|0.218|0.209|
>
> For frequently selected layers, the average noise ratios are also stable:
> Llama-3.1-8B (layers 15/16/17): 0.096 / 0.104 / 0.111;
> Qwen3-8B (layers 20/21/22): 0.111 / 0.123 / 0.110.
>
> We approximate this term away only to simplify the derivation and expose the dominant competition between **sink bias** and **structural aggregation**. If treated as a small residual instead of exactly zero, the qualitative conclusion is unchanged: the sink still consumes attention budget and contracts the structural component, while the residual is lower-order. We also tested removing this term at inference on Cycle task:
> |Model|Vanilla|SLASH|SLASH (remove noise)|
> |-|-|-|-:|
> |Llama-3.1-8B |0.660|0.680|0.686|
> |Qwen3-8B |0.395|0.905|0.838|
>
> This suggests that the residual is not the main driver, but it is also not always beneficial to remove it.
>
> **Q4/W3: Generality and benchmark scope.** SLASH is primarily about the **mechanism of how LLMs process explicit graph topology in serialized form**. We therefore focus on (i) pure graph computation tasks, where topology is the main signal, and (ii) molecular graphs, where structure remains important in a realistic setting. These settings let us evaluate the proposed mechanism most directly. As a small representative real-world check, we additionally evaluated a subset on **Cora**:
> |Method|Vanilla|SLASH|
> |-|-|-|
> |Llama-3.1-8B|0.900|**0.914**|
> |Qwen3-8B|0.329|**0.600**|
>
> This provides preliminary evidence that SLASH can also help beyond tasks where structural reasoning is central. At the same time, the current method depends on serialized graph descriptions with explicit edge structure, so it is not naturally suited to settings such as link prediction or heterogeneous / multi-edge-type graphs. Extending the current mechanism to richer graph formats is an important direction for future work.
>
> **Limitations.** For the limitation on compatibility with efficient attention implementations, please see our response to Reviewer `iUsP`. For scope limitation, this is exactly as discussed above.
>
> Thank you again for these precise and constructive questions.

---

> > ### Author Rebuttal · Reviewer_PHFv · 2026-04-02
> >
> > I found the rebuttal helpful and it addressed a substantial portion of my concerns. Some questions about scope and generalization remain, especially beyond serialized explicit-topology settings, but overall my assessment has improved and I am increasing my score from 3 to 4.

---

> > > ### Author Response · Authors · 2026-04-04
> > >
> > > Dear Reviewer PHFv,
> > >
> > > Thank you very much for the thoughtful follow-up and for your encouraging updated assessment. We sincerely appreciate your careful re-evaluation, and we are glad that our rebuttal helped address a substantial part of your concerns.
> > >
> > > To further address the remaining question on **scope and generalization**, we conducted an additional small representative evaluation based on **InstructGraph** test data on three task families. For each task, we used a subset of roughly 100 examples and adapted the serialized input format for SLASH. This was not included in our first-round response because, as we noted there, the original mechanism was developed for serialized graph descriptions with explicit edge structure and was therefore not originally designed for richer graph formats such as heterogeneous graphs. In this follow-up round, to probe this issue more directly, we implemented a simple adaptation for such settings: for heterogeneous graphs, we explicitly serialize the relation as an edge label in the form (i->j,r), which allows us to evaluate SLASH in these richer graph formats.
> > >
> > > | Task                | Dataset      | Model        | Vanilla |     SLASH |
> > > | ------------------- | ------------ | ------------ | ------: | --------: |
> > > | Node Classification | Cora         | Llama-3.1-8B |   0.900 | **0.914** |
> > > |                     |              | Qwen-8B      |   0.329 | **0.600** |
> > > | Link Prediction     | Wikidata5M   | Llama-3.1-8B |   0.230 | **0.260** |
> > > |                     |              | Qwen-8B      |   0.210 |     0.210 |
> > > | Graph QA            | PathQuestion | Llama-3.1-8B |   0.329 | **0.357** |
> > > |                     |              | Qwen-8B      |   0.357 | **0.386** |
> > >
> > > These results provide preliminary evidence that SLASH can also be useful in broader graph settings. More importantly, they further support the central perspective of our paper: SLASH tends to be helpful when topological structure plays an important role in the prediction or reasoning process, which is consistent with our mechanistic understanding of topology-aware signals inside LLMs. More broadly, we believe these findings help open up a broader perspective on how LLMs can be adapted more effectively for graph-structured data.
> > >
> > > Thank you again for the careful reading, constructive follow-up, and positive update.

---

### Official Review · Reviewer_HoLJ · 2026-03-13

**Soundness:** 3
**Presentation:** 3
**Significance:** 2
**Originality:** 2
**Overall Recommendation:** 5
**Confidence:** 4

**Summary:**

This work conducts a very interesting study of how LLMs internally process graphs and uncover spontaneous reconstruct graph topology..
It suggests to produce a “sawtooth” attention pattern aligned with the token level adjacency matrix.
The paper. suggests the use of SPLASH, a training free, plug and play attention sharpening method that reallocates attention away from the sink and toward topological structure, and so, help to avoid representation bottleneck arising from a fundamental conflict between the anisotropic bias acquired during language pre training and the isotropic information flow required for graph reasoning.

**Compliance With Llm Reviewing Policy:**

Affirmed.

**Final Justification:**

after reading the comments of the authors I adjust the grad

**Key Questions For Authors:**

- please Improve clarity on the research gap:
- Please explicitly state why existing methods fail and what key question remains unanswered
- Figure 4, that aims to examine RQ4 illustrates how SLASH corrects reasoning failures is missing deeper explanations.
- Discuss risks of over amplification.

**Limitations:**

RQ1: aimed to show that the method can significantly improves the performance, but it indicates that it is efficient only for general purpose LLMs and the gain is minimal for the domain-specific and fine-tuned cases

**Strengths And Weaknesses:**

The paper is well written and well motivated. It introduces a novel and compelling idea that can help to resolve some of the major limitations of current GNN based systems

In general. I liked the new proposed SPLASH methods and it looks like it provides a good direction to handle one of the major problems that limit current GNN based systems

---

> ### Author Rebuttal · Authors · 2026-03-31
>
> Dear Reviewer HoLJ,
>
> Thank you very much for the encouraging and supportive review. We truly appreciate your recognition of the paper as a **novel and compelling idea**, and especially your view that SLASH points to a **good direction** for addressing an important limitation in current graph reasoning systems. Your suggestions on clarifying the motivation, the mechanism, and the scope are also very helpful.
>
> **Q1: On the research gap.**
> Our central motivation is that existing Graph+LLM approaches mainly follow two paths: either they rely on external graph modules / fine-tuning, or they directly serialize graphs into text but largely treat the LLM as a black box. What remains unclear is **how LLMs internally process serialized graph topology**, and in particular why strong semantic models still struggle on structural graph reasoning. Our paper addresses this gap by showing that LLMs are not structurally blind: they already contain a latent topology-reconstruction signal (“sawtooth”), but this signal is diluted by the attention sink. This mechanistic diagnosis is exactly what motivates SLASH: instead of injecting an external graph encoder, it amplifies the model’s own latent structural understanding at inference time.
>
> **Q2: Why existing methods fail, and what remains unanswered.**
> The key issue is that existing methods can improve performance, but they usually do not explain **what internal mechanism suppresses structural reasoning once the graph is flattened into text**. Our work focuses on this missing question: if LLMs already possess latent structural reasoning ability, what prevents that ability from being fully expressed during serialized graph processing? Our answer is that the attention sink creates a bottleneck that competes with topology-aware local aggregation, and SLASH is designed directly from this diagnosis.
>
> **Q3: Deeper explanation of Figure 4.**
> This case study is intended to highlight the type of reasoning failure that SLASH corrects. In the vanilla case, the model answers “Yes” by producing a path that is not supported by the input graph, i.e., a structurally ungrounded reasoning trace. After sharpening, the model answers “No,” which is consistent with the actual connectivity. The point of Figure 4 is therefore not only that the final answer changes, but that SLASH helps the model’s reasoning stay better grounded in the graph structure itself by strengthening topology-aware aggregation relative to sink-dominated distraction.
>
> **Q4: On the risk of over-amplification.**
> This is an important point. In our framework, over-amplification corresponds to setting $\gamma$ too small, so that too much sink mass is removed and the structural component is over-sharpened. This can increase topological amplification, but at the cost of damaging model fidelity and destabilizing the representation. That is exactly why SLASH uses a calibrated intermediate $\gamma \in (0,1)$ rather than hard sink removal. Empirically, in most model-task pairs, performance drops sharply as $\gamma$ approaches 0, which is consistent with this over-amplification risk.
>
> **On the limitations.**
> We agree that the current results show the strongest gains on general-purpose LLMs, while the improvements on domain-specific / fine-tuned models are much smaller. Our interpretation is that this is not a contradiction to the method’s goal, but rather consistent with its mechanism: SLASH is designed to amplify latent structural signals that are already present but underused. When a model has already been specialized through graph-focused fine-tuning, part of this structural bias may already be internalized, so the room for additional improvement becomes naturally smaller. In this sense, the current evidence suggests that SLASH is especially valuable as a lightweight alternative to adaptation for general LLMs, while naturally leaving less room for gain once structural specialization has already been introduced through fine-tuning.
>
> Thank you again for the encouraging review and the very helpful suggestions.

---

> > ### Author Rebuttal · Reviewer_HoLJ · 2026-03-31
> >
> > On one hand, the authors answer my main concern, but on the other hand, considering my evaluation of how significant the paper is, I decided to keep my evaluation.
> >
> > I decided to adjust the grad

---

> > > ### Author Response · Authors · 2026-04-01
> > >
> > > Dear Reviewer HoLJ,
> > >
> > > Thank you again for your supportive evaluation. We are also very glad that our rebuttal addressed your concerns.
> > >
> > > On the significance aspect, one point we would like to share is that several reviewers independently highlighted a broader value of the work in similar terms. Reviewer `mYgY` described it as “a **pioneering understanding** of how LLMs work mechanistically on inputs with explicit relational/graph structure” and noted that “the insights from the paper can **open up areas** to explore how LLMs can be adapted more effectively for graph structured data.” Reviewer `PHFv` also emphasized the paper’s “**novel perspective** and **creative idea**” and “a **fresh mechanistic perspective**,” while Reviewer `iUsP` described the discovery of the sawtooth pattern as “a **compelling** visual and empirical finding.”
> > >
> > > We mention this because these comments, taken together, reflect the significance we hoped this work could have: a mechanistic perspective on Graph+LLM reasoning, together with a lightweight intervention, that may be useful for future research.
> > >
> > > Thank you again for your time and thoughtful comments. We would be very happy to discuss further if helpful.

---

### Official Review · Reviewer_mYgY · 2026-03-14

**Soundness:** 4
**Presentation:** 4
**Significance:** 3
**Originality:** 3
**Overall Recommendation:** 6
**Confidence:** 4

**Summary:**

This paper studies how LLMs internally process serialized graph structures which is how several recent works use LLMs for graph data processing. It then presents a quite interesting mechanistic finding- LLMs spontaneously reconstruct graph topology in their attention maps via a sawtooth pattern, but this latent structural signal is suppressed by the attention sink phenomenon. The paper formalizes this as a representation bottleneck building on the recent literature of attention sinks in LLMs. Based on the diagnosis, they propose SLASH, a training free plug-and-play method that redistributes attention away from the sink toward non sink tokens at inference time, amplifying the latent structural signal. Experiments on GraphInstruct and MolecularNet benchmarks across multiple LLMs show consistent performance improvements for open soure LLMs, with minimal effect on already fine-tuned models, hinting at how the proposed method can be an alternate LLM solution for graph tasks instead of finetuning.

**Compliance With Llm Reviewing Policy:**

Affirmed.

**Key Questions For Authors:**

included above

**Limitations:**

included above

**Strengths And Weaknesses:**

Strengths:
- The paper is very well written and to my reading a pioneering understanding of how LLMs work mechanistically on inputs with explicit relational/graph structure.
- Several recent works use LLMs to predict task labels on graphs and analyses there are often superficial. This work identifies a concrete internal phenomenon (the patterns with elaborate diagrams in the appendix) the provides a direct understanding of latent structural understanding.
- Although the proposed SLASH has multiple stages of preparation and also parameters which are model sensitive, it can still be used as plug and play on any LLMs.
- The insights from the paper can open up areas to explore how LLMs can be adapted more effectively for graph structured data, as LLMs occupy significant part in modern computing/AI stack.

Limitations and questions:
- To my best understanding, the framing of anisotropy vs. isotropy is imprecise and potentially misleading. The paper claims graph topology requires isotropic diffusion as the foundation of standard GNNs, but this directly contradicts the design of MPNNs and GATs which are explicitly anisotropic, assigning different weights to different neighbors. The paper even uses an implicit GAT as its own analogy in Section 4.2, which makes the isotropy claim internally inconsistent. The real conflict perhaps can be better characterized as the LLM's positional/causal sink bias conflicting with local neighborhood aggregation, where no single token should structurally dominate the isotropy framing could be misunderstood. I would be happy to discuss this point during the discussion period.
- There are obvious limitations some of which are included - the incompatibility with flash attention due the full matrix materialization, practical offline steps to prepare the inference operation (although these seem more sample efficient than finetuning), and truly new unseen tasks.
- typo in Figure 1: awtooth -> sawtooth.

---

> ### Author Rebuttal · Authors · 2026-03-31
>
> Dear Reviewer mYgY,
>
> Thank you very much for the highly positive and thoughtful review. We are especially encouraged that you highlighted several points that we also hoped would stand out as the paper’s core strengths:
> * a **pioneering mechanistic understanding** of serialized graph processing in LLMs
> * a concrete internal phenomenon—the **sawtooth pattern**—revealing latent structural understanding
> * a  **plug-and-play intervention** built directly on that diagnosis
>
> We are also particularly encouraged by your view that these insights may **open up new directions** for adapting LLMs more effectively to graph-structured data, which speaks directly to the broader significance we hoped this work could have. It is especially encouraging that closely related strengths—particularly the **novel perspective**, the **clear mechanistic explanation**, and the **accessibility of the intervention**—were also reflected in the reviews of Reviewers `iUsP`, `HoLJ`, and `PHFv`.
>
> **Q1: On the anisotropy vs. isotropy framing.**
> Thank you for this very precise point. You are right that our current wording can be made more accurate, and your comment helps us sharpen the conceptual framing.
>
> Our intended emphasis was on semantic anisotropy in LLMs: due to language pretraining, token representations tend to concentrate in a relatively narrow region of representation space, which in turn leads to a more directional semantic structure in processing. For serialized graph topology, however, the key requirement is different: the model must build representations through topology-aware local aggregation, rather than relying primarily on token-level semantic directionality.
>
> So the conflict we intended is better described as **semantic anisotropy vs. topology-aware local aggregation**, rather than saying that GAT itself is literally “isotropic.” As you pointed out, GAT/MPNN-style models are of course anisotropic in how they weight neighbors. Our intention was to describe the relevant LLM layers as exhibiting a GAT-like local message-passing behavior, while also noting that the pretrained semantic geometry of LLMs introduces an additional sink-dominated positional/semantic bias that can interfere with the local aggregation behavior needed for structural reasoning. In that sense, your formulation — “the LLM’s positional/causal sink bias conflicting with local neighborhood aggregation” — is essentially aligned with what we intended to express, and we will adopt a clearer wording along these lines in the paper.
>
> **Q2: On the limitations.**
> Thank you also for noting these practical boundaries. They help clarify the intended scope of SLASH: the current implementation requires attention-map access, the method includes a lightweight offline preparation stage, and broader transfer to truly unseen task families remains an important direction for future work.
>
> **Q3: On the typo in Figure 1.**
> Thank you for catching this. We will correct “awtooth” -> “sawtooth.”
>
> Thank you again for the very encouraging review and for the especially helpful correction to the conceptual framing. Your suggestion improves the clarity of the paper substantially.

---

> > ### Author Rebuttal · Reviewer_mYgY · 2026-04-04
> >
> > Thank you for your response and the clarifications to be made in the revised version.

---

> > > ### Author Response · Authors · 2026-04-04
> > >
> > > Dear Reviewer mYgY,
> > >
> > > Thank you very much for your thoughtful follow-up and for indicating that your concerns have been fully resolved. We sincerely appreciate your careful reading, your highly positive assessment of the paper, and your constructive suggestions.
> > >
> > > We are especially encouraged that the clarifications were helpful. Thank you again for your support and for the time and care you devoted to evaluating our work.

---

### Official Review · Reviewer_iUsP · 2026-03-23

**Soundness:** 2
**Presentation:** 3
**Significance:** 2
**Originality:** 3
**Overall Recommendation:** 3
**Confidence:** 4

**Summary:**

This paper addresses the challenge of Large Language Models (LLMs) processing structured data, specifically graph topologies, in serialized formats. The authors identify a "sawtooth" pattern in internal attention maps, suggesting that LLMs spontaneously reconstruct graph topology. They argue that this intrinsic capability is suppressed by the "attention sink"—a phenomenon where models allocate excessive attention to initial tokens. The paper introduces SLASH (Structural Attention SHarpening), a training-free, plug-and-play method that redistributes attention from the sink back to structural tokens. The method involves an offline calibration phase to identify topology-aware heads and an online phase for inference-time intervention. Experiments on GraphInstruct and MolecularNet show significant performance gains for general-purpose LLMs.

**Compliance With Llm Reviewing Policy:**

Affirmed.

**Final Justification:**

The author partially addressed my concerns. I agree that Slash is a lightweight tool compared to fine-tuning. However, the authors did not provide results regarding the model's general capabilities; I believe this is crucial for substantiating their comparison against fine-tuning. At this stage, the rebuttal content is insufficient for adjusting the score to positive.

**Key Questions For Authors:**

* Given the incompatibility with FlashAttention, did the authors measure the GPU memory overhead for larger graphs?
* Can the authors provide the results for fine-tuned models on non-aggregated inputs, similar to the analysis in Table 4?

**Limitations:**

yes

**Strengths And Weaknesses:**

### Strengths
* The discovery of the "sawtooth" pattern is a compelling visual and empirical finding.
* The authors provide a formal mathematical analysis of the "anisotropy-isotropy conflict" and offer a rigorous explanation of why the attention sink acts as a representation bottleneck for graph data.
* SLASH is training-free and introduces zero additional parameters, making it highly accessible and easy to integrate into existing inference pipelines.

### Weakness
* SLASH is incompatible with standard acceleration kernels like FlashAttention. This is a significant limitation for large-scale deployments or long-context graph processing.
* The identification of "topology-aware heads" and the observation of the "sawtooth" pattern are critically dependent on the proposed Source-Node Aggregation protocol. This protocol rearranges edges to ensure that those from the same source appear contiguously to create a "spatial basis" for observation. For real-world applications where graph data might be streamed or serialized in highly stochastic orders (e.g., large-scale web graphs), the reliance on a specific "spatial basis" might limit the method's broader applicability. While the authors provide a robustness analysis in Table 4, showing that SLASH still yields gains on non-aggregated inputs on Vanilla Instruct models, the authors should provide the results of fine-tuned models to support their claim.
* The results indicate that SLASH provides little benefit to models already fine-tuned for graph tasks (e.g., GraphWiz), as these models have likely already optimized their internal structural attention. Despite the clear benefits observed in general LLMs, it remains questionable whether SLASH provides gains for models trained on specialized graph data, especially for non-aggregated inputs.

---

> ### Author Rebuttal · Authors · 2026-03-31
>
> Dear Reviewer iUsP,
>
> Thank you for your careful reading and for highlighting three important aspects of our work: the novelty of the **sawtooth phenomenon**, the formal analysis of the **representation bottleneck**, and the accessibility of a **training-free, plug-and-play intervention**.
>
> **Q1/W1: FlashAttention incompatibility and GPU memory overhead.** This is an important practical consideration in the current implementation. In the current implementation, SLASH requires explicit access to the selected attention maps and is therefore not compatible with standard FlashAttention. To quantify the deployment cost, we measured peak GPU memory using Meta-Llama-3.1-8B-Instruct in 4-bit quantization, under the same inference setup, on both the largest real sample in our benchmark and a larger synthetic graph prompt for stress testing.
>
> The real sample contains 39 nodes, 387 edges, and 2416 tokens. Because this graph is still modest relative to the model context length, we additionally synthesized a larger graph with **100 nodes and 1141 edges**; under our current 2×24GB setup, this is the largest graph we can reliably run.
>
> | Setting | Tokens | Eager Peak (MB) | SLASH Eager Peak (MB) | FlashAttention2 Peak (MB) |
> | - | -: | -: | -: | -: |
> | Real sample | 2416 | 15855.5 | 15856.7 | 12161.3 |
> | Larger synthetic graph | 6942 | 36111.6 | 36111.7 | 13312.7 |
>
> These results show that the dominant deployment gap comes from forfeiting FlashAttention once full attention-map access is required, and that this gap becomes substantial as graph length grows. For the larger synthetic graph, peak memory increases from 13.3 GB with FlashAttention2 to 36.1 GB when the full attention maps are materialized. In other words, the main bottleneck is not the sharpening rule itself, but the need to access and edit the attention maps.
>
> We therefore agree that, in its current form, this limits scalability for long serialized graphs. At the same time, this limitation mainly stems from the current implementation requirement of full attention-map access, rather than from the sharpening rule itself. More flexible fused-attention backends that allow score modification at the kernel level may offer a path forward in future implementations; for example, designs such as FlexAttention suggest that this type of intervention may not be fundamentally out of reach, even though it is not supported in our current setup.
>
> **Q2/W2: Fine-tuned models under w/o Agg. inputs.** We agree that Source-Node Aggregation should be interpreted as an offline probing aid rather than a required online assumption. Its purpose is to provide a spatial basis that makes topology-aware heads easier to identify during the offline screening stage. Following your suggestion, we additionally evaluated fine-tuned graph models under **w/o Agg.** inputs:
>
> | Model               | Task  | Vanilla (w/o Agg.) | SLASH (w/o Agg.) |
> | ------------------- | ----- | -----------------: | ---------------: |
> | GraphWiz-LLaMA2-7B  | Cyc.  |              0.857 |            0.871 |
> |                     | Bip.  |              0.860 |            0.860 |
> |                     | Ham.  |              0.606 |            0.609 |
> | GraphWiz-Mistral-7B | Cyc.  |              0.903 |            0.894 |
> |                     | Bip.  |              0.729 |            0.729 |
> |                     | Ham.  |              0.257 |            0.274 |
> | MolecularGPT        | BBBP  |              0.430 |            0.430 |
> |                     | Tox21 |              0.490 |            0.505 |
>
> These additional results are consistent with the interpretation in the main paper: SLASH is most beneficial for general-purpose LLMs that already contain latent structural signals but do not use them efficiently, whereas **graph-specialized / graph-finetuned models** show much smaller changes because fine-tuning likely already partially internalizes, or compensates for, the relevant structural bias. Importantly, under **w/o Agg.** inputs, the fine-tuned models remain mostly neutral to mildly helpful rather than exhibiting any strong dependence on aggregated serialization.
>
> Together with the w/o Agg. results already reported for general-purpose LLMs in the main paper, these new results strengthen our intended interpretation that Source-Node Aggregation is an offline diagnostic/probing device for exposing topology-aware heads, not a required condition for applying SLASH at test time. This is also consistent with our original Table 4 observation that online sharpening remains effective even when the input edge order is not aggregated.
>
> Thank you again for these constructive suggestions.

---

> > ### Author Rebuttal · Reviewer_iUsP · 2026-04-02
> >
> > I would like to thank the authors for their detailed responses. But I believe addressing the concerns mentioned above requires a significant update to the paper.
> >
> > * The lack of a scalable implementation makes the current version feel like a "toy" setup that does not yet meet the requirements for deployment in real-world LLM contexts. Addressing this would require significant technical updates.
> >
> > * Besides, the experimental results show that SLASH provides only marginal gains for models that have already been fine-tuned on graph domains (e.g., GraphWiz variants). This raises a significant question regarding the work: if the gold standard for graph reasoning is fine-tuning (which also preserves model generalizability when done correctly), the practical niche for a training-free intervention that offers minimal improvement over SOTA models is narrow. Most critically, the impact of SLASH on the model’s general abilities (e.g., Mathematics, Coding, or standard NLP benchmarks like MMLU) remains unexamined. The authors' analysis of the Graph-SST2 task shows that SLASH can be counterproductive and degrade performance when the task relies on semantic understanding rather than pure topology. Without evidence that SLASH does not significantly degrade general capabilities, its utility is questionable. A better solution for a general-purpose LLM might be fine-tuning on a mixture of graph and general reasoning data, which strengthens graph understanding while maintaining overall performance. Validating SLASH against this common practice would require a massive update to the experimental section.

---

> > > ### Author Response · Authors · 2026-04-03
> > >
> > > Dear Reviewer iUsP,
> > >
> > > Thank you again for the careful follow-up. We believe the remaining concerns mainly relate to (i) the deployment maturity of the current implementation, and (ii) the practical niche of SLASH relative to fine-tuning and to broader workloads.
> > >
> > > **(1) On scalability and implementation maturity.**
> > > The main contribution of this paper is a **mechanistic understanding of how LLMs process serialized graph topology**: we identify a concrete internal phenomenon, show that it reflects latent structural processing, and validate this diagnosis with a targeted inference-time intervention. We believe the present evidence is sufficient to support this contribution in the studied regime.
> > >
> > > As clarified in our rebuttal, the current implementation requires explicit attention-map access, and we provided memory measurements to make this implementation boundary concrete. We therefore view the remaining concern as primarily about the scalability boundary of the current implementation, rather than about whether the paper’s central mechanistic claim is supported in the studied setting. As noted in the rebuttal, increasingly flexible accelerated attention designs also suggest a practical route for deploying flexible attention variants of this kind in future implementations.
> > >
> > > **(2) On practical niche, fine-tuning, and broader workloads.**
> > > We would also like to clarify the intended role of SLASH. We do not see SLASH and graph-specific fine-tuning as serving the same role. Fine-tuning is a valid and often effective route, but it relies on additional supervision and retraining to inject or strengthen graph-specific behavior. Our contribution is to show that **general-purpose LLMs already contain latent topology-related capability**, and that part of this underused capability can be better utilized at inference time without retraining.
> > >
> > > Under this view, SLASH has value as a lightweight, plug-and-play tool for topology-centric settings where retraining is undesirable or unavailable. The smaller gains on graph-specialized / graph-finetuned models are therefore consistent with the mechanism rather than contradictory to it: once graph-specific behavior has already been partially internalized through fine-tuning, the remaining headroom for an additional structural intervention naturally becomes smaller.
> > >
> > > The Graph-SST2 result is useful precisely because it marks the boundary of this setting. When a task depends mainly on semantic understanding rather than graph topology, SLASH is simply not intended to be activated. In broader workloads, it can be straightforwardly **disabled** when structural reasoning is not central to the task, leaving the base model unchanged.
> > >
> > > Overall, we believe the follow-up concerns mainly sharpen the scope and positioning of SLASH as a targeted intervention for topology-centric settings. They mainly raise valid questions about scalable implementation and the practical niche relative to fine-tuning, but the paper’s central contribution remains the same: identifying a concrete mechanistic phenomenon in how LLMs process serialized graph topology, and validating that diagnosis through a targeted training-free intervention.
> > >
> > > Thank you again for the thoughtful follow-up.

---

### Decision · Program_Chairs · 2026-04-30

**Decision:**

Accept (regular)

**Comment:**

This paper investigates how LLMs process serialized graph topology, identifying a "sawtooth" attention pattern that aligns with the token-level adjacency matrix.  The authors show that the attention sink phenomenon dilutes this signal, and based on this diagnosis, the authors propose SLASH, a training-free inference-time intervention that redistributes attention away from sink tokens. Reviewers appreciated the novelty of the mechanistic finding and the elegance of a plug-and-play solution requiring no additional parameters (mYgY, PHFv, HoLJ).  Primary concerns centered on incompatibility with FlashAttention limiting scalability (iUsP), limited evaluation scope beyond serialized explicit-topology settings (PHFv, iUsP), and marginal gains on already fine-tuned graph models (iUsP, HoLJ). The authors' rebuttal provided some new memory benchmarks, additional evaluations and ablations on design choices, which fully resolved concerns for two reviewers (mYgY, HoLJ) and partially addressed those of the others. While one reviewer (iUsP) maintained a weak reject citing deployment maturity and narrow practical niche relative to fine-tuning, reviewers generally valued the meaningful mechanistic contribution with a clean intervention and leaned towards accepting this paper at this time. I encourage the authors to iterate on the main points of feedback—particularly broader evaluation in non-serialized-graph settings for the final version.